# Adversarial Diffusion
# for Robust Reinforcement Learning

**Daniele Foffano**[*]
Division of Decision and Control Systems
KTH, Royal Institute of Technology
`foffano@kth.se`

**Alessio Russo**
Faculty of Computing and Data Sciences
Boston University
`arusso2@bu.edu`

**Alexandre Proutiere**
Division of Decision and Control Systems
KTH, Royal Institute of Technology
`alepro@kth.se`

## Abstract

Robustness to modeling errors and uncertainties remains a central challenge in reinforcement learning (RL). In this work, we address this challenge by leveraging diffusion models to train robust RL policies. Diffusion models have recently gained popularity in model-based RL due to their ability to generate full trajectories "all at once", mitigating the compounding errors typical of step-by-step transition models. Moreover, they can be conditioned to sample from specific distributions, making them highly flexible. We leverage conditional sampling to learn policies that are robust to uncertainty in environment dynamics. Building on the established connection between Conditional Value at Risk (CVaR) optimization and robust RL, we introduce Adversarial Diffusion for Robust Reinforcement Learning (AD-RRL). AD-RRL guides the diffusion process to generate worst-case trajectories during training, effectively optimizing the CVaR of the cumulative return. Empirical results across standard benchmarks show that AD-RRL achieves superior robustness and performance compared to existing robust RL methods.

## 1 Introduction

Reinforcement Learning (RL) has produced agents that surpass human-level performance in various domains [29, 55, 46, 45, 30]. However, the same policies are notoriously sensitive to small dynamics changes, sensor noise, or hardware mismatch, all of which can cause dramatic performance collapse. In safety–critical domains such as robotics, finance, or healthcare—where collecting new data is expensive, risky, or legally restricted—robustness to modeling errors is at least as important as maximizing nominal reward.

Model-based RL improves sample efficiency by learning a world-model and planning within it, but faces two key robustness obstacles: (i) compounding errors, which accumulate over long horizons [56, 8]; and (ii) the Sim2Real gap, where controllers that succeed in simulation fail after minor real-world deviations [41, 6]. Compounding error occurs in autoregressive models, where the model predicts one step ahead and then is fed its own prediction back: the state predicted at time $t$ is used to predict the state at $t+1$. Small one-step errors accumulate, the trajectory moves away from reality, and performance degrades. The Sim2Real gap arises because, even with a highly accurate simulator that minimizes unrealistic artifacts, simulated physics can never perfectly replicate reality. This

---

[*]Corresponding author

39th Conference on Neural Information Processing Systems (NeurIPS 2025).

discrepancy leads to reduced real-world performance due to inherent modeling inaccuracies. To overcome these challenges, RL algorithms should be made more robust by optimizing not only the expected return but also the performance under adverse or uncertain dynamics.

Recently, diffusion models have been proposed to mitigate compounding errors by generating entire trajectories rather than predicting one step at a time [38, 17]. While this reduces error accumulation, diffusion models remain imperfect: the trajectories they generate may deviate from real-world dynamics. As a result, transferring policies learned in simulation to the real world remains challenging. Despite recent progress, diffusion-based RL methods often struggle to maintain robustness when deployed in environments with unseen or perturbed dynamics.

Adversarial and risk-sensitive approaches have been explored to enhance robustness against model errors. These methods introduce worst-case perturbations during planning [35, 37], or optimize Conditional Value at Risk (CVaR) objectives, which have been shown to improve resilience to both reward variability and model inaccuracies [34, 5]. In this work, we show that diffusion models and CVaR-based approaches can be seamlessly integrated to complement each other. We propose Adversarial Diffusion for Robust Reinforcement Learning (AD-RRL), a novel algorithm that combines the strengths of diffusion models and CVaR-based robustness. By leveraging trajectory-level generation to mitigate compounding errors and incorporating risk-aware objectives, AD-RRL enhances the adaptability and robustness of RL agents to modeling mismatches and environmental uncertainty. More precisely, we make the following contributions.

(a) We present Adversarial Diffusion (AD), a guided diffusion model that for a given policy, generates trajectories that are challenging for the agent and result in relatively low rewards. These trajectories are either rare in the current environment or originate from unexplored regions of the domain. We show that by learning from such adversarial scenarios, the agent can improve its robustness to modeling errors. To generate these trajectories, we leverage the CVaR framework, applied to trajectory rewards, and demonstrate how guided diffusion can be used to efficiently implement this objective. This mechanism forms the foundation of AD.

(b) Building on this, we introduce AD-RRL, our RL algorithm that integrates AD within the Advantage Actor-Critic (A2C) framework. AD-RRL significantly enhances the agent's adaptability and robustness. We empirically evaluate AD-RRL across multiple environments from the Gym/MuJoCo suite, showing that it achieves superior robustness to modeling errors. In transfer scenarios involving variations in physics parameters, AD-RRL consistently outperforms state-of-the-art baselines.

## 2    Related Work

**Model-Based Reinforcement Learning.**    In Model-Based Reinforcement Learning, the agent uses a model to generate new data, through which it is possible to plan further without interacting with the environment. This is essential for settings where collecting new data is impossible, illegal or dangerous. The parametric approach has received a lot of attention thanks to the constant improvements of function approximators, such as Deep Neural Networks [32, 7, 18, 16, 53]. Recently, Variational Auto Encoders [19] and Transformers [54] have seen many successful applications as powerful models for environment dynamics [28, 39, 42, 12, 11], leading to state-of-the-art methods in terms of sample efficiency and performance [13]. These methods rely on bootstrapping to generate trajectory samples. The state prediction generated by the model is fed again as input to the model to predict the next state. As a result, these methods introduce two sources of error: one coming from an imperfect model and one from the input of the model always being wrong, except for the first timestep. The sum of these errors is commonly known as the Compounding Error problem of Model-Based methods. Multi-step prediction solutions have been proven effective even before the introduction of Diffusion models, for example by learning $H$ models to look $H$ steps into the future [3]. It goes without saying that this approach results in a much higher learning complexity with respect to single-model approaches. Only recently, with Diffusion models becoming more popular, we have seen the rise of more efficient multi-step Model-Based RL methods [38, 17].

**Diffusion Models in Reinforcement Learning.**    Diffusion models are inspired by non-equilibrium thermodynamics [47, 15], defining data generation as an iterative denoising process. Beyond being powerful function approximators, they also offer a natural way to condition the data generation process on labels [10]. Recently, diffusion models have gained significant attention in the RL

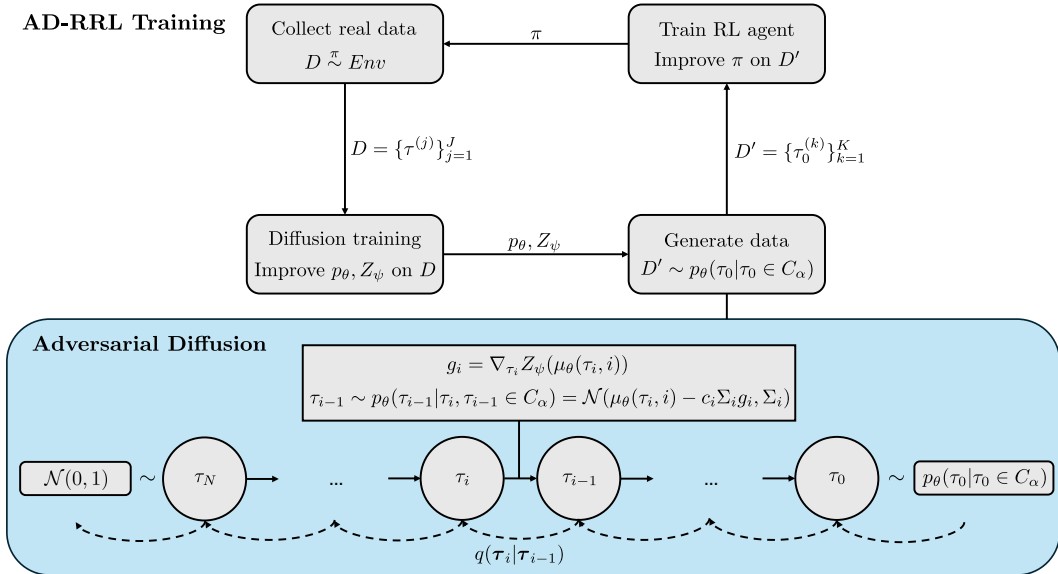

Figure 1: A high-level overview of AD-RRL. Following a Dyna-like structure [49], the algorithm iteratively: samples trajectories from the real environment with a given policy $\pi$, improves the Diffusion models on the collected data, uses an Adversarially Guided Diffusion model (Section 4) to generate challenging synthetic trajectories which in turn are used to improve $\pi$. The loop is repeated until convergence to the optimal policy $\pi^\star$.

community. They have been used to model system dynamics, generating trajectory segments by predicting either states [1, 58], actions [4, 22], or both [17, 24]. Guidance techniques can further refine trajectory generation by conditioning the process on value estimates, promoting high-expected-reward sequences. Additionally, diffusion models have been employed for policy modeling [56, 14] and value function approximation [26]. Most research on diffusion models in RL has focused on the offline setting. In this paper, we shift the focus to the online case, building on PolyGRAD—an online, Dyna-style Model-Based RL method that uses diffusion for modeling dynamics [38]. While prior work primarily uses conditioning to generate high-reward trajectories, we take the opposite approach. Our goal is to generate challenging trajectories—those that are underexplored or unlikely—so the agent can learn a more robust policy, better suited to handling changes in dynamics and modeling errors.

**Robust Reinforcement Learning.**   Classical RL can struggle to generalize when test environments deviate from training due to model errors or shifts. Robust RL addresses this by accounting for uncertainty in actions, states, and dynamics. One line of work regularizes transition probabilities within a defined uncertainty set [9, 20, 25]. This kind of methods, despite being theoretically sound and robust, do not scale well to more complex environment.

A well-known approach to tackling complex robust RL problems while maintaining theoretical guarantees is to frame the optimization problem as a two-player game [31]. In this framework, two players are trained iteratively to solve a maximin optimization problem: the primary agent aims to maximize the expected cumulative reward, while an adversarial agent attempts to minimize it by introducing disturbances. For instance, in Robust Adversarial Reinforcement Learning (RARL) [35], the adversary applies external forces to disturb the environment's dynamics. Max-min TD3 (M2TD3) [50] follows a similar strategy, solving a maximin problem to maximize the expected reward under worst-case scenarios within an uncertainty set. In Noisy Action Robust MDPs [51], the adversary perturbs the agent's actions, while in State Adversarial MDPs [48], the adversary introduces perturbations to the state, resulting in a Partially Observable MDP formulation.

Several Robust RL algorithms use CVaR to constrain their optimization problems [57]. For instance, CVaR-PPO is an extension of PPO [44] solving a risk-sensitive constrained optimization problem that constrains the CVaR to a given threshold.

Finally, we have algorithms using Domain Randomization (DR) [52], where the agent maximizes the expected return on average, over a predefined uncertainty set for some given environment parameters. These classes of methods have been proven very effective in domains such as robotics [23]. However, they do not aim to be robust to the worst-case scenarios, and might fall short when tested on environments outside of their training distribution.

## 3 Background and Problem Statement

### 3.1 Markov Decision Processes and Reinforcement Learning

Consider a Markov Decision Process (MDP) $M = \langle S, A, P, r, \gamma \rangle$, where $S$ and $A$ are the state and action spaces, respectively, $P(\cdot|s, a)$ is the transition probability function, $r(\cdot|s, a)$ is the reward function and $\gamma$ is the discount factor. By interacting with the MDP, a Reinforcement Learning agent is able to collect sequences of states, actions and rewards, forming trajectories $\boldsymbol{\tau} = (s_0, a_0, r_0, \dots, s_H, a_H, r_H)$. The objective of the Reinforcement Learning agent is to learn an optimal policy $\pi^\star$ maximizing the policy value $v_\pi(s) = \mathbb{E}_\pi[\sum_{i=0}^\infty \gamma^i r_{t+i+1}|s_t = s]$. In this paper, we consider a model-based RL setting, where we use a diffusion model to approximate the distribution of trajectories under a given policy. Specifically, if $p^\pi$ denotes the true distribution of the trajectories $\boldsymbol{\tau}$ under policy $\pi$, the diffusion model samples trajectories with distribution $p_{\boldsymbol{\theta}}$ close to $p^\pi$. We adopt a Dyna-style approach [49], where the diffusion model and the policy are iteratively updated: the policy is improved using data collected from the model, while the model is improved using samples gathered from the target environment using the learned policy.

### 3.2 Robust RL through the Conditional Value at Risk.

We now discuss Conditional Value at Risk (CVaR) and its connection to Robust RL.

**Conditional Value at Risk.** When learning policies robust to modeling errors, a framework commonly used is the one of Conditional Value at Risk. We define the return of a trajectory $\boldsymbol{\tau}$ by $Z(\boldsymbol{\tau}) = \sum_{t=0}^H \gamma^t r_t$, where $r_t$ is the reward obtained at time $t$ in this trajectory[2]. Under a policy $\pi$, $Z(\boldsymbol{\tau})$ is a random variable with cdf $F$. The Value-at-Risk (VaR) of $Z$ at confidence level $\alpha \in (0, 1)$ corresponds to its $\alpha$ quantile:

$$\text{VaR}_\alpha^\pi(Z) = \max\{z|F(z) \leq \alpha\}. \tag{1}$$

The Conditional Value-at-Risk (CVaR) of $Z$ is then defined as the expected value of $Z$ on the lower $\alpha$-portion of its distribution

$$\text{CVaR}_\alpha^\pi(Z) = \mathbb{E}_\pi[Z|Z \leq \text{VaR}_\alpha^\pi(Z)]. \tag{2}$$

**CVaR dual formulation and its connection to robustness to modeling errors.** An alternative way of defining CVaR stems from its dual formulation [2, 40]:

$$\text{CVaR}_\alpha^\pi(Z) = \min_{\xi \in \mathcal{U}_{\text{CVaR}}^{\alpha,\pi}} \mathbb{E}_{\boldsymbol{\tau} \sim p^\pi}[\xi(\boldsymbol{\tau})Z(\boldsymbol{\tau})], \tag{3}$$

where $\xi$ acts as a perturbation of the return $Z$. This perturbation belongs to the set $\mathcal{U}_{\text{CVaR}}^{\alpha,\pi}$, called the *risk envelope* and defined as

$$\mathcal{U}_{\text{CVaR}}^{\alpha,\pi} := \left\{ \xi : \forall \boldsymbol{\tau}, \xi(\boldsymbol{\tau}) \in \left[0, \frac{1}{\alpha}\right], \mathbb{E}_{\boldsymbol{\tau} \sim p^\pi}[\xi(\boldsymbol{\tau})] = 1 \right\}. \tag{4}$$

(3) states that the CVaR of $Z$ can be defined as its expected value under a worst-case perturbed distribution.

In RL, optimizing a CVaR objective introduces robustness to model misspecification. This is exactly because of the dual form of CVaR, where the trajectory distribution is distorted by an adversarial density $\xi(\boldsymbol{\tau})$. CVaR optimization in this case equals maximizing the worst-case discounted reward when adversarial perturbations are budgeted over the whole trajectory rather than at each time step [5]. The connection between CVaR and robustness to modeling errors is well established in the RL field [34, 37, 35], and the dual formulation is at the core of our method, as explained in Section 4 and Appendix B.

---

[2]To avoid cluttering, we write $Z$ instead of $Z(\boldsymbol{\tau})$ unless it is required to avoid misunderstandings.

### 3.3 Diffusion Models

In this work, we adopt a model-based approach to learn robust policies. To achieve this, we harness the efficiency of diffusion processes to learn a parameterized model $p_{\boldsymbol{\theta}}$ of the trajectory distribution. This model allows us to sample trajectories $\boldsymbol{\tau}$ as if they were generated by the true MDP, enabling policy training on these synthetic trajectories.

Diffusion models generate data by progressively refining noisy inputs through an iterative denoising process, $p_{\boldsymbol{\theta}}(\boldsymbol{\tau}_{i-1}|\boldsymbol{\tau}_i)$. This process reverses the forward diffusion, $q(\boldsymbol{\tau}_i|\boldsymbol{\tau}_{i-1})$, which gradually corrupts real data by adding random noise. Each step of the denoising process is typically parameterized as a Gaussian distribution

$$p_{\boldsymbol{\theta}}(\boldsymbol{\tau}_{i-1}|\boldsymbol{\tau}_i) = \mathcal{N}(\mu_{\boldsymbol{\theta}}(\boldsymbol{\tau}_i, i), \boldsymbol{\Sigma}_i), \tag{5}$$

with learned mean and fixed covariance matrices, both depending on the diffusion step $i$.

The denoising process is formulated as

$$p_{\boldsymbol{\theta}}(\boldsymbol{\tau}_{0:N}) = p(\boldsymbol{\tau}_N) \prod_{i=1}^{N} p_{\boldsymbol{\theta}}(\boldsymbol{\tau}_{i-1}|\boldsymbol{\tau}_i), \tag{6}$$

where $p(\boldsymbol{\tau}_N) \approx \mathcal{N}(\mathbf{0}, \boldsymbol{I})$ and $\boldsymbol{\tau}_0$ is the real (i.e., noiseless) trajectory. The parameters $\boldsymbol{\theta}$ are learned by optimizing the variational lower bound on the negative log likelihood:

$$\boldsymbol{\theta}^{\star} = \arg\min_{\boldsymbol{\theta}} \mathbb{E}_{\boldsymbol{\tau}_0}[-\log p_{\boldsymbol{\theta}}(\boldsymbol{\tau}_0)], \tag{7}$$

where $p_{\boldsymbol{\theta}}(\boldsymbol{\tau}_0) = \int p_{\boldsymbol{\theta}}(\boldsymbol{\tau}_{0:N}) \mathrm{d}\boldsymbol{\tau}_{1:N}$.

**Guided diffusion.** A classifier $p(y|\boldsymbol{\tau}_0)$ adding information about the sample to be reconstructed (e.g., the optimality of the trajectory) can enhance the generative performance of the diffusion model [10]

$$p_{\boldsymbol{\theta}}(\boldsymbol{\tau}_0|y) \propto p_{\boldsymbol{\theta}}(\boldsymbol{\tau}_0)p(y|\boldsymbol{\tau}_0). \tag{8}$$

By leveraging the classifier's gradient, we can guide the denoising process toward generating samples that align more closely with the classifier's predictions. This method, called Classifier-Guided Diffusion, generates trajectory samples according to

$$p_{\boldsymbol{\theta}}(\boldsymbol{\tau}_{i-1}|\boldsymbol{\tau}_i, y) = \mathcal{N}(\mu_{\boldsymbol{\theta}}(\boldsymbol{\tau}_i, i) + \boldsymbol{\Sigma}_i \boldsymbol{g}_i, \boldsymbol{\Sigma}_i) \tag{9}$$

where $\boldsymbol{g}_i = \nabla_{\boldsymbol{\tau}} \log p(y|\boldsymbol{\tau})|_{\boldsymbol{\tau}=\mu_{\boldsymbol{\theta}}(\boldsymbol{\tau}_i, i)}$.

### 3.4 Problem statement

We consider a model-based Reinforcement Learning setting. For a given policy $\pi$, we learn a model $p_{\boldsymbol{\theta}}$ of the distribution of the corresponding trajectories. Our goal is to use $p_{\boldsymbol{\theta}}$ to improve the policy and its robustness to modeling errors.

We formulate the problem of learning a robust policy using the following optimization problem:

$$\pi_{\alpha}^{\star} = \arg\max_{\pi} \mathrm{CVaR}_{\alpha}^{\pi}(Z) \tag{10}$$

$$= \arg\max_{\pi} \min_{\xi \in \mathcal{U}_{\mathrm{CVaR}}^{\alpha, \pi}} \mathbb{E}_{\boldsymbol{\tau} \sim p^{\pi}}[\xi(\boldsymbol{\tau})Z(\boldsymbol{\tau})]. \tag{11}$$

(10) describes the objective to obtain the policy that maximizes the return on the worst $\alpha$-percentile of the trajectories, in terms of cumulative return. However, directly sampling trajectories from this worst $\alpha$-percentile is challenging. In our approach, we leverage the dual definition of CVaR, solving instead the double optimization problem described in (11). The problem can be seen as a game where an adversarial agent $\xi$ is perturbing the trajectories distribution under a given policy $\pi$. We model this distribution via a diffusion model $p_{\boldsymbol{\theta}}$, which allows us to leverage guiding techniques. We introduce *adversarial guiding*, a method that steers the diffusion process toward sampling trajectories that minimize the expected return for the agent. Because the adversarial guide actively seeks to reduce return, the generated trajectories naturally fall within the worst $\alpha$-percentile. We formally demonstrate that the resulting adversarially guided diffusion models can be adapted to actually sample from the worst $\alpha$-percentile. We also empirically validate our approach.

# 4 Adversarially Guided Diffusion Models

In this section, we consider a fixed policy $\pi$, and for notational convenience, we drop the corresponding superscripts. We explain below how to efficiently generate adversarial trajectories, sampled from the set of trajectories $C_\alpha := \{\boldsymbol{\tau} : Z(\boldsymbol{\tau}) \leq \text{VaR}_\alpha(Z)\}$.

**Sampling suboptimal trajectories.** To steer the diffusion process towards the set $C_\alpha$ we need to define the proper guidance classifier, as in (8). We start from the definition of $\text{CVaR}_{\alpha,p_{\boldsymbol{\theta}}}(Z)$ given in (2). The index $p_{\boldsymbol{\theta}}$ indicates that the trajectory $\boldsymbol{\tau}$ from which the return is computed is generated using the diffusion model $p_{\boldsymbol{\theta}}$. We have:

$$\text{CVaR}_{\alpha,p_{\boldsymbol{\theta}}}(Z) = \mathbb{E}_{\tau_0 \sim p_{\boldsymbol{\theta}}}[Z(\boldsymbol{\tau}_0)|\boldsymbol{\tau}_0 \in C_\alpha], \tag{12}$$

$$= \int Z(\boldsymbol{\tau}_0)p_{\boldsymbol{\theta}}(\boldsymbol{\tau}_0|\boldsymbol{\tau}_0 \in C_\alpha)d\boldsymbol{\tau}_0, \tag{13}$$

$$= \min_{\xi \in \mathcal{U}_{\text{CVaR}}^{\alpha,\pi}} \int Z(\boldsymbol{\tau}_0)\xi(\boldsymbol{\tau}_0)p_{\boldsymbol{\theta}}(\boldsymbol{\tau}_0)d\boldsymbol{\tau}_0, \tag{14}$$

where the last equality follows from the dual definition of CVaR presented in (3). To steer the generating process towards trajectories from $C_\alpha$, we can use the classifier $p_{\boldsymbol{\theta}}(\boldsymbol{\tau}_0 \in C_\alpha|\boldsymbol{\tau}_0)$, since $p_{\boldsymbol{\theta}}(\boldsymbol{\tau}_0|\boldsymbol{\tau}_0 \in C_\alpha) \propto p_{\boldsymbol{\theta}}(\boldsymbol{\tau}_0)p_{\boldsymbol{\theta}}(\boldsymbol{\tau}_0 \in C_\alpha|\boldsymbol{\tau}_0)$.

Notice also that if we define $\xi^\star(\boldsymbol{\tau}_0)$ as the solution to the minimization problem in (14), i.e., the bounded trajectory perturbation minimizing the cumulative reward under the dynamics $p_{\boldsymbol{\theta}}$, we have

$$\xi^\star(\boldsymbol{\tau}_0)p_{\boldsymbol{\theta}}(\boldsymbol{\tau}_0) = p_{\boldsymbol{\theta}}(\boldsymbol{\tau}_0|\boldsymbol{\tau}_0 \in C_\alpha) \propto p_{\boldsymbol{\theta}}(\boldsymbol{\tau}_0)p_{\boldsymbol{\theta}}(\boldsymbol{\tau}_0 \in C_\alpha|\boldsymbol{\tau}_0).$$

In other words, weighting the distribution with the classifier $p_{\boldsymbol{\theta}}(\boldsymbol{\tau}_0 \in C_\alpha \mid \boldsymbol{\tau}_0)$ is equivalent, up to a proportionality constant, to weighting $p_{\boldsymbol{\theta}}(\tau_0)$ according to an adversarial perturbation $\xi^\star$. We can hence think of applying a guided diffusion to implement this perturbation. However, the set $C_\alpha$ is not known. To address this limitation, and following the approach of [21], we introduce a smooth approximation of $p_{\boldsymbol{\theta}}(\boldsymbol{\tau}_0 \in C_\alpha \mid \boldsymbol{\tau}_0)$, namely $\exp(-c_0 \sum_{t=1}^H \gamma^t r_t)$ for some constant $c_0 > 0$. This approximation is intuitively reasonable, as it biases the generation process toward trajectories with lower cumulative rewards.

In the following two subsections, we describe how this guided diffusion can be implemented and how it influences the diffusion process. We also discuss how to tune the guided diffusion to ensure that the resulting adversarial perturbation $\xi$ remains within the risk envelope $\mathcal{U}_{\text{CVaR}}^{\alpha,\pi}$ defined in (4).

## 4.1 Perturbed diffusion model

We use the classifier $p_{\boldsymbol{\theta}}(\boldsymbol{\tau}_i \in C_\alpha|\boldsymbol{\tau}_i)$ so that the trajectories $\boldsymbol{\tau}_i$ generated at every step $i$ of the diffusion process belong to the set $C_\alpha$. We assume that $p_{\boldsymbol{\theta}}(\boldsymbol{\tau}_i \in C_\alpha|\boldsymbol{\tau}_i) \approx \exp\left(-c_i \sum_{t=1}^H \gamma^t r_t^{(i)}\right)$ for some value $c_i > 0$ as we did for $\boldsymbol{\tau}_0$. In the last approximation, the reward $r_t^{(i)}$ represents the reward collected at time $t$ in trajectory $\boldsymbol{\tau}_i$ for the $i$-th step of the diffusion process.

As a slight extension of the guided diffusion principle presented in Section 3.3, we establish the following result (essentially obtained by applying (9) with $y = \{\boldsymbol{\tau}_i \in C_\alpha\}$).

**Lemma 4.1.** *Assume that the denoising process is Gaussian, that is (5) holds. Assume that for all $i \in [N]$, the approximation $p_{\boldsymbol{\theta}}(\boldsymbol{\tau}_i \in C_\alpha|\boldsymbol{\tau}_i) = \exp\left(-c_i \sum_{t=1}^H \gamma^t r_t^{(i)}\right)$ holds. Then, we can sample trajectories from $p_{\boldsymbol{\theta}}(\boldsymbol{\tau}_0|\boldsymbol{\tau}_0 \in C_\alpha)$ using diffusion steps of the form:*

$$p_{\boldsymbol{\theta}}(\boldsymbol{\tau}_{i-1}|\boldsymbol{\tau}_i, \boldsymbol{\tau}_{i-1} \in C_\alpha) = \mathcal{N}(\mu_{\boldsymbol{\theta}}(\boldsymbol{\tau}_i, i) - c_i \boldsymbol{\Sigma}_i \boldsymbol{g}_i, \boldsymbol{\Sigma}_i), \tag{15}$$

*where $\boldsymbol{g}_i = \nabla_{\boldsymbol{\tau}} Z(\mu_{\boldsymbol{\theta}}(\boldsymbol{\tau}_i, i))$ for $i \in [N]$.*

The lemma is proved in Appendix A for completeness. The conditional sampling procedure induces the following perturbed model:

$$\bar{p}_{\boldsymbol{\theta}}(\boldsymbol{\tau}_0) = p_{\boldsymbol{\theta}}(\boldsymbol{\tau}_0|\boldsymbol{\tau}_0 \in C_\alpha) \propto \int \prod_{i=1}^N p_{\boldsymbol{\theta}}(\boldsymbol{\tau}_{i-1}|\boldsymbol{\tau}_i, \boldsymbol{\tau}_{i-1} \in C_\alpha)p(\boldsymbol{\tau}_N)d\boldsymbol{\tau}_{1:N}. \tag{16}$$

We refer to this sampling procedure as an *Adversarially Guided Diffusion Model*.

## 4.2 Selecting $c_1, \ldots, c_N$

Note that the Adversarially Guided Diffusion Model depends on the constants $c_1, \ldots, c_N$, and recall that the resulting perturbation must lie within the risk envelope defined in (4). In the following, we establish conditions on these constants to ensure this requirement is satisfied. To that end, we first show in Appendix B that our model $\bar{p}_{\boldsymbol{\theta}}$ admits a product-form representation:

**Lemma 4.2.** *The Adversarially Guided Diffusion Model can be expressed as* $\bar{p}_{\boldsymbol{\theta}}(\boldsymbol{\tau}_0) = \xi(\boldsymbol{\tau}_0)p_{\boldsymbol{\theta}}(\boldsymbol{\tau}_0)$, *where* $\xi(\boldsymbol{\tau}_0) = \frac{\int \xi(\boldsymbol{\tau}_{0:N})p_{\boldsymbol{\theta}}(\boldsymbol{\tau}_{0:N})d\boldsymbol{\tau}_{1:N}}{p_{\boldsymbol{\theta}}(\boldsymbol{\tau}_0)}$ *and where* $\xi(\boldsymbol{\tau}_{0:N}) := \prod_{i=1}^{N} \xi(\boldsymbol{\tau}_i, \boldsymbol{\tau}_{i-1})$ *with*

$$\xi(\boldsymbol{\tau}_i, \boldsymbol{\tau}_{i-1}) := \exp\left(-\frac{1}{2}(2c_i \boldsymbol{D}_i^T \boldsymbol{g}_i + c_i^2 \boldsymbol{g}_i^T \boldsymbol{\Sigma} \boldsymbol{g}_i)\right), \tag{17}$$

*and* $\boldsymbol{D}_i := (\boldsymbol{\tau}_{i-1} - \mu_{\boldsymbol{\theta}}(\boldsymbol{\tau}_i, i))$.

Note that by definition (since $\bar{p}_{\boldsymbol{\theta}}$ is a distribution), we have that $\mathbb{E}_{\boldsymbol{\tau} \sim p_{\boldsymbol{\theta}}}[\xi(\boldsymbol{\tau}_0)] = 1$. Hence, we just need to verify that $\xi(\boldsymbol{\tau}_0) \leq 1/\alpha$ for all $\boldsymbol{\tau}_0$ to ensure that $\xi$ belongs to the risk envelope. We define $\rho$ such that the trajectories $\boldsymbol{\tau}_i$ lie in a bounded space $C = \{\boldsymbol{\tau}_i : ||\boldsymbol{\tau}_i||_\infty \leq \rho\}$ and such that $||\mu_{\boldsymbol{\theta}}(\boldsymbol{\tau}_i, i)||_\infty < \rho$. In the following proposition, proved in Appendix C, we provide conditions on $c_1, \ldots, c_N$ so that this holds.

**Proposition 4.3.** *For all* $i \in [N]$, *let* $\eta_i(\alpha, N) \geq 0$ *such that* $\prod_{i=1}^{N} \eta_i(\alpha, N) = \frac{1}{\alpha}$.
*(a) When for all* $i \in [N]$,

$$c_i \leq \min\left(\sqrt{\frac{2\log\eta_i(\alpha, N)}{\boldsymbol{g}_i^T \boldsymbol{\Sigma}_i \boldsymbol{g}_i}}, \frac{\rho - ||\mu_{\boldsymbol{\theta}}(\boldsymbol{\tau}_i, i)||_\infty}{||\boldsymbol{\Sigma}_i \boldsymbol{g}_i||_\infty}\right), \tag{18}$$

*then we have: for all* $\boldsymbol{\tau}_0$, $\xi(\boldsymbol{\tau}_0) \leq 1/\alpha$.
*(b) Let* $i \in [N]$. *Assume that* $\boldsymbol{\Sigma}_i$ *is a diagonal matrix with* $(\boldsymbol{\Sigma}_i)_{jj} \in [0, 1)$. *Assume* $\eta_i(\alpha, N) = \left(\frac{1}{\alpha}\right)^{\frac{1}{N}}$. *Then, for* $N$ *large enough, (18) holds as soon as* $c_i \leq \sqrt{\frac{2\log\eta_i(\alpha, N)}{\boldsymbol{g}_i^T \boldsymbol{\Sigma}_i \boldsymbol{g}_i}}$.

Since our diffusion model uses a cosine noise schedule as in [33], we have that for all $i \in [N]$ $\boldsymbol{\Sigma}_i = \beta_i \boldsymbol{I}$ with $\beta_i \in [0, 1)$, so we can set $c_i = \sqrt{\frac{2\log\eta_i(\alpha, N)}{\boldsymbol{g}_i^T \boldsymbol{\Sigma}_i \boldsymbol{g}_i}}$ to ensure that $\xi$ belongs to the risk envelope.

*Remark* 4.4. Note that in our analysis, we have assumed for simplicity that the states and the actions were a one-dimensional vector, so that trajectories become Gaussian vectors. We can extend the analysis to the case where states and actions are multidimensional at the expense of considering trajectories as Gaussian matrices. Refer to Appendix D for details.

# 5 Algorithms

We now introduce Adversarial Diffusion for Robust Reinforcement Learning (AD-RRL)[3], which alternates between model improvement and policy improvement steps (see Algorithm 1). AD-RRL leverages the adversarial conditional sampling discussed in the previous section to sample trajectories in the worst $\alpha$-percentile in terms of return. Our approach is primarily inspired by PolyGRAD [38] and Diffuser [17].

We adopt the common assumption that the policy follows a Gaussian distribution over the action space, parameterized by $\mu_\omega(s)$ and $\sigma_\omega(s)$. The policy is deployed in the real environment to collect new data, which is then used to train both the dynamics model $\bar{p}_{\boldsymbol{\theta}}$ and the cumulative reward function $Z_\phi$. Following the standard approach in Dyna-like algorithms [49], we generate synthetic trajectories using our learned models (Algorithm 2). These trajectories are then used to train the policy via an on-policy Reinforcement Learning algorithm.

---

[3]The official implementation for AD-RRL can be found on GitHub: `https://github.com/danielefoffano/AD-RRL`

**Algorithm 1** Adversarial Diffusion for Robust Reinforcement Learning (AD-RRL)

1: **Input:** environment, $E$;
2: **Initialize:** policy, $\pi_\omega$; adversarial denoising model, $\bar{p}_\theta$; cumulative reward function $Z_\phi$; data buffer, $\mathcal{D}$; training iterations $M$

3: **for** $m = 1, \ldots, M$ **do**
4:      Sample $\tau \sim E$ using $\pi_\omega$, add $\tau$ to $\mathcal{D}$
5:      Improve $\bar{p}_\theta$, $Z_\phi$ on $\mathcal{D}$     ▷ Algorithm 3
6:      Sample $\{\hat{\tau}\} \sim \bar{p}_\theta$     ▷ Algorithm 2
7:      Improve $\pi_\omega$ on $\{\hat{\tau}\}$ using RL
8: **end for**

**Algorithm 2** Adversarial Diffusion Trajectory Sampling

1: **Input:** adversarial denoising model $\bar{p}_\theta$;
2: reward model $Z_\phi$; buffer $\mathcal{D}$; level $\alpha$

3: $\hat{\tau}_N \sim \mathcal{N}(\mathbf{0}, \mathbf{I})$
4: $s_0 \sim \mathcal{D}$
5: **for** $i = N, \ldots, 1$ **do**
6:      set $\hat{s}_0 \leftarrow s_0$ in $\hat{\tau}_i$
7:      $c_i = \sqrt{2 \log \eta_i(\alpha, N)/\boldsymbol{g}_i^T \boldsymbol{\Sigma}_i \boldsymbol{g}_i}$
8:      $\hat{\tau}_{i-1} \sim \mathcal{N}\big(\mu_\theta(\hat{\tau}_i, i) - c_i \boldsymbol{\Sigma}_i \boldsymbol{g}_i, \ \boldsymbol{\Sigma}_i\big)$
9: **end for**
10: **return** $\hat{\tau}_0$

Algorithm 3 illustrates the training procedure for both our diffusion models and is presented in Appendix E, alongside additional implementation details. The pseudocode provided is simplified. In reality, the diffusion model consists of a noise prediction function $\epsilon_\theta(\hat{\tau}_i, i)$ from which the mean is computed in closed form [15]. This model is trained using the following objective function (derived from (7))

$$\mathcal{L}(\boldsymbol{\theta}) = \mathbb{E}_{i, \epsilon, \tau_0}[||\epsilon - \epsilon_\theta(\tau_i, i)||^2],$$

where $i \sim \mathcal{U}(\{1, \ldots, N\})$ is the diffusion process step, $\epsilon \sim \mathcal{N}(0, 1)$ is the target noise and $\tau_i$ is the trajectory $\tau_0 \sim \mathcal{D}$ after $i$ steps of the *forward* diffusion process adding noise $\epsilon$. We update $\boldsymbol{\theta}$ $K$ times, each time by randomly sampling the step $i$. The model $Z_\phi$ is trained to predict the cumulative reward of the trajectory samples $\tau_i$.

Both the adversarial diffusion model $\bar{p}_\theta$ and the cumulative reward function $Z_\phi$ are used to sample adversarially generated trajectories in the worst $\alpha-$percentile. At every step of the diffusion process, we perform inpainting by substituting a real starting state $s_0$ into the generated noisy trajectory $\hat{\tau}_i$. We then proceed to compute $c_i$ according to (18) and the gradient $\boldsymbol{g}_i = \nabla_s Z_\phi$. Notice that the gradient is taken with respect to $s$, so we only adversarially corrupt the states of the trajectory. To ensure that the generated actions are consistent with the generated states, we use the PolyGRAD diffusion guidance method [38], which generates a sequence of actions guided by the gradient of the policy $\pi_\omega$.

## 6 Experiments

In this section, we empirically evaluate how robust our method is. During training, the agent interacts with a fixed instance of the environment. At test time, we alter key physics-related parameters and assess the agent's performance against both robust and non-robust baselines. Our experiments are conducted on several optimal control tasks from the MuJoCo suite: InvertedPendulum, Reacher, Hopper, HalfCheetah, and Walker. All agents are trained in the default MuJoCo/OpenAI Gym environment (fixed physics), for 1.5M steps. Additional results are provided in Appendix F.

**Baseline methods.** We evaluate AD-RRL against several state-of-the-art baselines for robust reinforcement learning:

(a) Domain Randomization (DR) [52], widely used in robotics [23, 27], improves policy generalization by maximizing expected return over a distribution of dynamics. However, it does not explicitly account for worst-case or lower-percentile outcomes. We implement DR using PPO and refer to the resulting method as DR-PPO.

(b) Max-Min TD3 (M2TD3) [50] frames robustness as a minimax optimization problem, training an actor-critic model to maximize performance under the worst-case dynamics sampled from a predefined uncertainty set.

(c) CVaR-PPO (CPPO) [57] augments Proximal Policy Optimization with a CVaR constraint, leading to a policy-gradient algorithm that explicitly controls the policy's risk.

Additionally, we compare AD-RRL to other baselines in RL.

(d) PolyGRAD [38], a diffusion-based model that our work builds upon, generates synthetic trajectories via policy-guided diffusion and trains policies in an online model-based setting. It improves sample efficiency but lacks explicit robustness to adverse dynamics.

(e) TRPO [43] and PPO [44], two strong model-free baselines, are also included for comparison. TRPO constrains policy updates using a KL-divergence trust region, while PPO employs a clipped surrogate objective for improved computational efficiency.

**Robustness under varying physical parameters.** To verify the robustness of AD-RRL, we vary several physical parameters of the environment at test time. For Hopper and Cheetah, we vary body mass, ground friction and environment gravity. For Walker, we modify friction and mass. For Reacher, we vary all the actuators' gears (i.e., the torque produced by the actions). For InvertedPendulum, we change the cart mass, the pole mass and environment gravity.

In Figure 2, we plot the return under the different algorithms and for selected environments and varying parameters. Additional plots are provided in Appendix F.2. In most environments, AD-RRL consistently outperforms both robust and non-robust baselines. PPO and TRPO appear surprisingly stable, which is likely a consequence of the well-tuned Stable-Baselines3 implementations—but are still matched or surpassed by AD-RRL. At the same time, AD-RRL consistently outperforms both DR-PPO and M2TD3, demonstrating greater stability and achieving higher cumulative rewards across all environments.

Furthermore, a direct comparison with PolyGRAD (the foundation of our algorithm) highlights that our modifications significantly improve performance under diverse test-time conditions, enhancing robustness to large parameter shifts and model misspecifications. This can be clearly seen in Figure 2d or Figure 2g. In the Reacher environment (Figure 2i) the difference in performance is less evident, but our model still performs consistently better or on par with the baselines. It is also clear from Figures 2a and 2b that while PolyGRAD achieves slightly better performance on the nominal environment (as observed in Table 1), it sacrifices robustness under perturbed conditions.

For some environments—see for example Figure 2a and Figure 4b (presented in Appendix F), AD-RRL performance degrades for extreme changes in the modified parameter (but it remains better than other algorithms). We hypothesize that this is because our model is generating challenging trajectories which are nonetheless plausible under the agent policy and environment dynamics. Extreme changes in the environment physics do not reflect these constraints, and relevant trajectories might not be generated often.

**Performance on the nominal (training) environments.** Table 1 reports the final episode returns (mean $\pm$ one standard error) for five MuJoCo continuous-control tasks. Best results are highlighted in bold. The results are obtained on the training environment, with the nominal physics parameters. AD-RRL attains the best mean return on four of the five domains, substantially outperforming the other baselines, showing that our risk-sensitive training does not trade nominal optimality for robustness. Only on Hopper, PolyGRAD performs better than AD-RRL, but the margin falls within overlapping confidence intervals. On the easy Inverted Pendulum task, multiple methods (AD-RRL, DR-PPO, PPO) reach the maximum score of 1000, as expected.

| | Hopper | Cheetah | Walker | Reacher | InvertedPendulum |
|---|---|---|---|---|---|
| AD–RRL | $3280.23 \pm 13.83$ | $\mathbf{4126.11 \pm 246.96}$ | $\mathbf{4357.80 \pm 187.37}$ | $\mathbf{-3.97 \pm 0.13}$ | $\mathbf{1000.00 \pm 0.00}$ |
| PolyGRAD | $\mathbf{3346.99 \pm 52.39}$ | $3879.16 \pm 626.40$ | $3489.48 \pm 456.70$ | $-4.48 \pm 0.13$ | $\mathbf{1000.00 \pm 0.00}$ |
| M2TD3 | $361.73 \pm 13.71$ | $3117.16 \pm 55.34$ | $2948.03 \pm 598.77$ | $-21.28 \pm 5.75$ | $634.76 \pm 192.46$ |
| CPPO | $2595.64 \pm 298.35$ | $2173.30 \pm 422.97$ | $2164.30 \pm 510.60$ | $-6.06 \pm 0.32$ | $979.85 \pm 20.15$ |
| DR–PPO | $2315.90 \pm 482.17$ | $2429.46 \pm 558.93$ | $2385.19 \pm 589.49$ | $-15.80 \pm 1.44$ | $\mathbf{1000.00 \pm 0.00}$ |
| PPO | $2998.90 \pm 432.28$ | $2408.20 \pm 546.33$ | $1894.03 \pm 349.06$ | $-5.17 \pm 0.57$ | $\mathbf{1000.00 \pm 0.00}$ |
| TRPO | $3270.27 \pm 273.04$ | $2014.91 \pm 539.64$ | $3090.80 \pm 267.79$ | $-6.22 \pm 0.85$ | $960.60 \pm 39.40$ |

Table 1: Return on the training environment (nominal physics parameters) for MuJoCo continuous-control tasks.

**Sample efficiency.** The learning curves in Figure 3 (see Appendix F.1) show that AD-RRL reaches higher or matching final performance with the same number of samples as the baselines, and in several cases converges faster. Hence, our adversarially guided diffusion not only preserves (or improves) performance on the training environment (with nominal physics parameters), but also matches the sample efficiency of state-of-the-art model-based and model-free alternatives. In Appendix F.3, we perform an ablation study comparing AD-RRL trained on 1.5M samples against the Model-Free

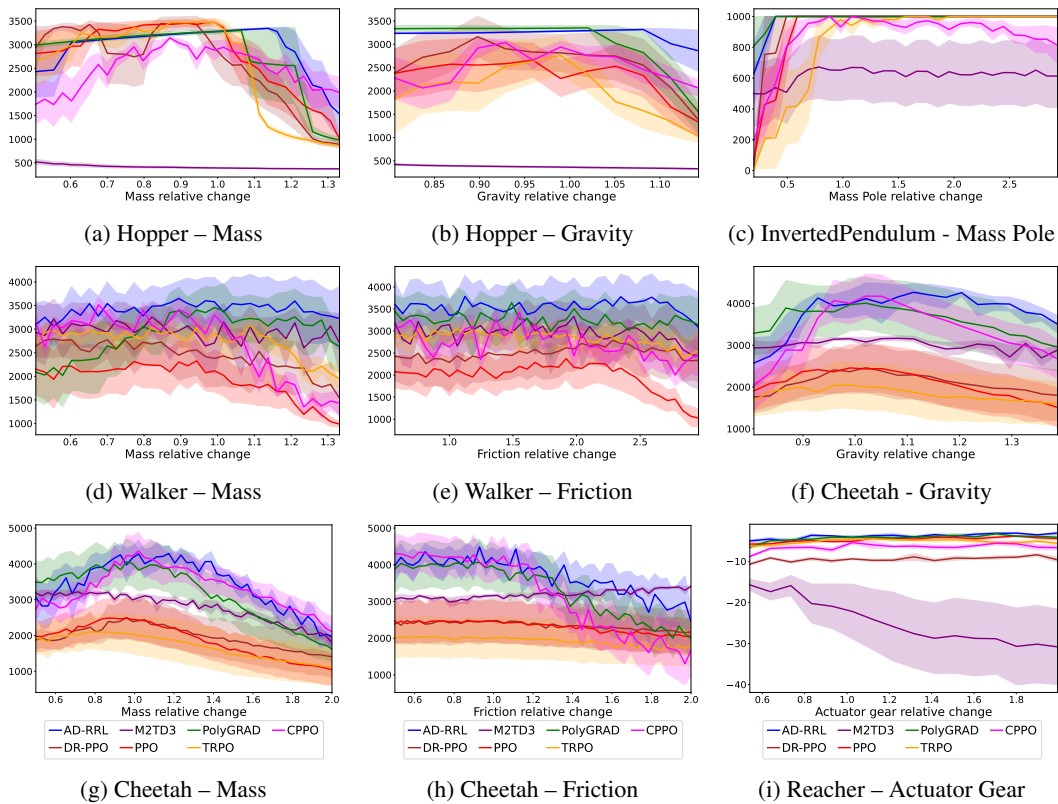

Figure 2: Average return across variations in selected physics parameters. Shaded regions indicate $\pm$ one standard error.

Baselines trained on 3M samples. This way we allow the model-free methods to compensate for their lower sample efficiency by leveraging their computational efficiency. Overall, Figures 5 and 6 show that even when model-free baselines are given twice as many samples, AD-RRL remains on average better or on par across the proposed tasks.

# 7   Conclusion and Future Work

In this work we introduced AD-RRL, a novel approach to robust RL. AD-RRL is based on Adversarial Diffusion (AD), a diffusion model that can sample adversarial trajectories by leveraging the Conditional Value at Risk (CVaR) framework. AD enables agents to learn from adversarial scenarios that are either rare or unexplored in the environment. We demonstrated that AD-RRL, based on this diffusion model, significantly enhances the robustness of RL agents in the presence of modeling errors. Through empirical evaluation on multiple Gym/MuJoCo environments, we showed that AD-RRL outperforms current state-of-the-art robust RL methods.

AD relies on a specific strategy for guiding the diffusion process, and exploring alternative guidance methods presents a promising avenue for future work. Potential directions include (i) modifying the overall diffusion objective beyond the current CVaR framework, and (ii) enhancing the diffusion model architecture or algorithms to reduce computational overhead.

# 8 Acknowledgments

The computations were enabled by resources provided by the National Academic Infrastructure for Supercomputing in Sweden (NAISS), partially funded by the Swedish Research Council through grant agreement no. 2022-06725. We also thank Adam Miksits for the invaluable support provided in the development of the numerical results.

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

# Contents

# A  Adversarially Guided Diffusion Models: Proofs of results from Section 4.1

*Proof of Lemma 4.1.* For a sufficiently smooth function $r$, the conditional distribution $\bar{p}_{\boldsymbol{\theta}}(\boldsymbol{\tau}_i|\boldsymbol{\tau}_{i+1})$ can be approximated using a Gaussian. Following [47, Appendix C] (see also [10]) we know that

$$p_{\boldsymbol{\theta}}(\boldsymbol{\tau}_{i-1}|\boldsymbol{\tau}_i, \boldsymbol{\tau}_{i-1} \in C_\alpha) \approx \mathcal{N}(\mu_{\boldsymbol{\theta}}(\boldsymbol{\tau}_i, i) + \Sigma_i \boldsymbol{h}_i, \Sigma_i)$$

where $\boldsymbol{h}_i = \nabla_{\boldsymbol{\tau}} \log p_{\boldsymbol{\theta}}(\boldsymbol{\tau} \in C_\alpha|\boldsymbol{\tau})|_{\boldsymbol{\tau}=\mu_{\boldsymbol{\theta}}(\boldsymbol{\tau}_i, i)}$.

Since we assume that the approximation $p_{\boldsymbol{\theta}}(\boldsymbol{\tau}_i \in C_\alpha|\boldsymbol{\tau}_i) = \exp\left(-c_i \sum_{t=1}^H \gamma^t r_t^{(i)}\right)$ holds, we get that

$$
\begin{aligned}
\boldsymbol{h}_i &= \nabla_{\boldsymbol{\tau}} \log p(\boldsymbol{\tau} \in C_\alpha|\boldsymbol{\tau})|_{\boldsymbol{\tau}=\mu_{\boldsymbol{\theta}}(\boldsymbol{\tau}_i, ti)} \\
&= -c_i \sum_{t=1}^T \nabla_{\boldsymbol{s_t},\boldsymbol{a_t}} r(s_t, a_t)|_{(s_t, a_t)=\mu_{\boldsymbol{\theta}}^t(\boldsymbol{\tau}_i, i)} \\
&= -c_i \nabla_{\boldsymbol{\tau}} Z(\boldsymbol{\tau})|_{\boldsymbol{\tau}=\mu_{\boldsymbol{\theta}}(\boldsymbol{\tau}_i, i)},
\end{aligned}
$$

where $\mu_{\boldsymbol{\theta}}^t(\boldsymbol{\tau}_i, i)$ is the $t$-th state-action pair of $\mu_{\boldsymbol{\theta}}(\boldsymbol{\tau}_i, i)$. Substituting, we get

$$p_{\boldsymbol{\theta}}(\boldsymbol{\tau}_{i-1}|\boldsymbol{\tau}_i, \boldsymbol{\tau}_{i-1} \in C_\alpha) = \mathcal{N}(\mu_{\boldsymbol{\theta}}(\boldsymbol{\tau}_i, i) - c_i \Sigma_i \boldsymbol{g}_i, \Sigma_i), \tag{19}$$

where $\boldsymbol{g}_i = \nabla_{\boldsymbol{\tau}} Z(\boldsymbol{\tau})|_{\boldsymbol{\tau}=\mu_{\boldsymbol{\theta}}(\boldsymbol{\tau}_i, i)}$. $\qquad\square$

*Proof of Equation* (16). As mentioned earlier, to sample from $p_{\boldsymbol{\theta}}(\boldsymbol{\tau}_0|\boldsymbol{\tau}_0 \in C_\alpha)$, we multiply each intermediate distribution in the diffusion process by $r_i(\boldsymbol{\tau}_i)$, with $r_i(\boldsymbol{\tau}_i) = \exp(-c_i \sum_{t=1}^H \gamma^t r_t^{(i)})$, where the notation $r_t^{(i)}$ refers to the $t$-th reward in $\boldsymbol{\tau}_i$ for the $i$-th diffusion step. This means that the corresponding modified distribution $\bar{p}_{\boldsymbol{\theta}}$ satisfies in the intermediate diffusion step $i$:

$$\bar{p}_{\boldsymbol{\theta}}(\boldsymbol{\tau}_i) = \frac{1}{\tilde{Z}_i} r_i(\boldsymbol{\tau}_i) p_{\boldsymbol{\theta}}(\boldsymbol{\tau}_i), \tag{20}$$

where $\tilde{Z}_i$ is the normalizing constant. Next, we use the same strategy as that used in [47] to determine the diffusion process $\bar{p}_{\boldsymbol{\theta}}(\boldsymbol{\tau}_i|\boldsymbol{\tau}_{i+1})$. Note first that:

$$\bar{p}_{\boldsymbol{\theta}}(\boldsymbol{\tau}_i) = \int \bar{p}_{\boldsymbol{\theta}}(\boldsymbol{\tau}_i|\boldsymbol{\tau}_{i+1}) \bar{p}_{\boldsymbol{\theta}}(\boldsymbol{\tau}_{i+1}) \mathrm{d}\boldsymbol{\tau}_{i+1}.$$

Plugging (20), the previous condition can be rewritten as

$$p_{\boldsymbol{\theta}}(\boldsymbol{\tau}_i) = \int \bar{p}_{\boldsymbol{\theta}}(\boldsymbol{\tau}_i|\boldsymbol{\tau}_{i+1}) \frac{\bar{Z}_i}{\bar{Z}_{i+1}} \frac{r_{i+1}(\boldsymbol{\tau}_{i+1})}{r_i(\boldsymbol{\tau}_i)} p_{\boldsymbol{\theta}}(\boldsymbol{\tau}_{i+1}) \mathrm{d}\boldsymbol{\tau}_{i+1}. \tag{21}$$

However, we know that $p_{\boldsymbol{\theta}}$ also satisfies:

$$p_{\boldsymbol{\theta}}(\boldsymbol{\tau}_i) = \int p_{\boldsymbol{\theta}}(\boldsymbol{\tau}_i|\boldsymbol{\tau}_{i+1}) p_{\boldsymbol{\theta}}(\boldsymbol{\tau}_{i+1}) \mathrm{d}\boldsymbol{\tau}_{i+1}.$$

This implies that (21) holds if:

$$\bar{p}_{\boldsymbol{\theta}}(\boldsymbol{\tau}_i|\boldsymbol{\tau}_{i+1}) = p_{\boldsymbol{\theta}}(\boldsymbol{\tau}_i|\boldsymbol{\tau}_{i+1}) \frac{\bar{Z}_{i+1} r_i(\boldsymbol{\tau}_i)}{\bar{Z}_i r_{i+1}(\boldsymbol{\tau}_{i+1})}.$$

Now defining the normalization constant $\bar{Z}_i(\boldsymbol{\tau}_{i+1}) = \frac{\bar{Z}_{i+1}}{\bar{Z}_i r_{i+1}(\boldsymbol{\tau}_{i+1})}$, we get

$$\bar{p}_{\boldsymbol{\theta}}(\boldsymbol{\tau}_i|\boldsymbol{\tau}_{i+1}) = \frac{1}{\bar{Z}_i(\boldsymbol{\tau}_{i+1})} p_{\boldsymbol{\theta}}(\boldsymbol{\tau}_i|\boldsymbol{\tau}_{i+1}) r_i(\boldsymbol{\tau}_i).$$

We conclude that $p_{\boldsymbol{\theta}}(\boldsymbol{\tau}_i|\boldsymbol{\tau}_{i+1}, \boldsymbol{\tau}_i \in C_\alpha) \propto \tilde{p}_{\boldsymbol{\theta}}(\boldsymbol{\tau}_i|\boldsymbol{\tau}_{i+1})$. Therefore, we have shown that:
$p_{\boldsymbol{\theta}}(\boldsymbol{\tau}_0|\boldsymbol{\tau}_0 \in C_\alpha) = \bar{p}_{\boldsymbol{\theta}}(\boldsymbol{\tau}_0),$

$$
\begin{aligned}
&= \int \bar{p}_{\boldsymbol{\theta}}(\boldsymbol{\tau}_0|\boldsymbol{\tau}_1) \bar{p}_{\boldsymbol{\theta}}(\boldsymbol{\tau}_1) \mathrm{d}\boldsymbol{\tau}_1, \\
&= \int \bar{p}_{\boldsymbol{\theta}}(\boldsymbol{\tau}_0|\boldsymbol{\tau}_1) \cdots \bar{p}_{\boldsymbol{\theta}}(\boldsymbol{\tau}_{N-1}|\boldsymbol{\tau}_N) p_{\boldsymbol{\theta}}(\boldsymbol{\tau}_N) \mathrm{d}\boldsymbol{\tau}_1, \ldots, \boldsymbol{\tau}_N, \\
&\propto \int p_{\boldsymbol{\theta}}(\boldsymbol{\tau}_0|\boldsymbol{\tau}_1, \boldsymbol{\tau}_0 \in C_\alpha) \cdots p_{\boldsymbol{\theta}}(\boldsymbol{\tau}_{N-1}|\boldsymbol{\tau}_N, \boldsymbol{\tau}_{N-1} \in C_\alpha) p_{\boldsymbol{\theta}}(\boldsymbol{\tau}_N) \mathrm{d}\boldsymbol{\tau}_1, \ldots, \boldsymbol{\tau}_N.
\end{aligned}
$$

$\qquad\square$

# B Adversarial guide as a multiplicative perturbation: Proof of Lemma 4.2

*Proof of Lemma 4.2.* Let's define a denoising diffusion model $p_{\boldsymbol{\theta}}(\boldsymbol{\tau}_{i-1}|\boldsymbol{\tau}_i)$, and a perturbed denoising step of the form $p_{\boldsymbol{\theta}}(\boldsymbol{\tau}_{i-1}|\boldsymbol{\tau}_i, \boldsymbol{\tau}_{i-1} \in C_\alpha) = \mathcal{N}(\mu_{\boldsymbol{\theta}}(\boldsymbol{\tau}_i, i) - c_i \boldsymbol{\Sigma}_i \boldsymbol{g}_i, \boldsymbol{\Sigma}_i)$. Since the two distributions are Gaussians with known mean and covariance matrices, we have

$$p_{\boldsymbol{\theta}}(\boldsymbol{\tau}_{i-1}|\boldsymbol{\tau}_i, \boldsymbol{\tau}_{i-1} \in C_\alpha) = \mathcal{N}(\mu_{\boldsymbol{\theta}}(\boldsymbol{\tau}_i, i) - c_i \boldsymbol{\Sigma}_i \boldsymbol{g}_i, \boldsymbol{\Sigma}_i)$$

$$= K \exp\left(-\frac{1}{2}(\boldsymbol{D}_i + c_i \boldsymbol{\Sigma}_i \boldsymbol{g}_i)^T \boldsymbol{\Sigma}_i^{-1} (\boldsymbol{D}_i + c_i \boldsymbol{\Sigma}_i \boldsymbol{g}_i)\right)$$

$$= K \exp\left(-\frac{1}{2}(\boldsymbol{D}_i^T \boldsymbol{\Sigma}_i \boldsymbol{D}_i + 2c_i \boldsymbol{D}_i^T \boldsymbol{g}_i + c_i^2 \boldsymbol{g}_i^T \boldsymbol{\Sigma}_i \boldsymbol{g}_i)\right)$$

$$= K \exp\left(-\frac{1}{2} \boldsymbol{D}_i^T \boldsymbol{\Sigma}_i \boldsymbol{D}_i\right) \exp\left(-\frac{1}{2}(2c_i \boldsymbol{D}_i^T \boldsymbol{g}_i + c_i^2 \boldsymbol{g}_i^T \boldsymbol{\Sigma}_i \boldsymbol{g}_i)\right)$$

$$= \mathcal{N}(\boldsymbol{\tau}_{i-1}|\mu_{\boldsymbol{\theta}}(\boldsymbol{\tau}_i, i), \boldsymbol{\Sigma}_i) \exp\left(-\frac{1}{2}(2c_i \boldsymbol{D}_i^T \boldsymbol{g}_i + c_i^2 \boldsymbol{g}_i^T \boldsymbol{\Sigma}_i \boldsymbol{g}_i)\right)$$

$$= \xi(\boldsymbol{\tau}_i, \boldsymbol{\tau}_{i-1}) p_{\boldsymbol{\theta}}(\boldsymbol{\tau}_{i-1}|\boldsymbol{\tau}_i)$$

where $K = \frac{1}{(2\pi)^{d/2}|\boldsymbol{\Sigma}_i|^{1/2}}$, $\boldsymbol{D}_i = (\boldsymbol{\tau}_{i-1} - \mu_{\boldsymbol{\theta}}(\boldsymbol{\tau}_i, i))$, and $\xi(\boldsymbol{\tau}_i, \boldsymbol{\tau}_{i-1}) = \exp\left(-\frac{1}{2}(2c_i \boldsymbol{D}_i^T \boldsymbol{g}_i + c_i^2 \boldsymbol{g}_i^T \boldsymbol{\Sigma} \boldsymbol{g}_i)\right)$. $\mathcal{N}(\boldsymbol{\tau}_{i-1}|\mu_{\boldsymbol{\theta}}(\boldsymbol{\tau}_i, i), \boldsymbol{\Sigma}_i)$ is the density of the Gaussian distribution of $\boldsymbol{\tau}_{i-1}$, with mean $\mu_{\boldsymbol{\theta}}(\boldsymbol{\tau}_i, i)$ and covariance matrix $\boldsymbol{\Sigma}_i$.

If we define $\xi(\boldsymbol{\tau}_{0:N}) = \prod_{i=1}^{N} \xi(\boldsymbol{\tau}_i, \boldsymbol{\tau}_{i-1})$, we have

$$p_{\boldsymbol{\theta}}(\boldsymbol{\tau}_{0:N}|\boldsymbol{\tau}_0 \in C_\alpha) = \xi(\boldsymbol{\tau}_{0:N}) p(\boldsymbol{\tau}_{0:N})$$

So we can define

$$\bar{p}_{\boldsymbol{\theta}}(\boldsymbol{\tau}_0) = p_{\boldsymbol{\theta}}(\boldsymbol{\tau}_0|\boldsymbol{\tau}_0 \in C_\alpha) = \int p_{\boldsymbol{\theta}}(\boldsymbol{\tau}_{0:N}|\boldsymbol{\tau}_0 \in C_\alpha) = 1)\mathrm{d}\boldsymbol{\tau}_{1:N}$$

$$= \int \xi(\boldsymbol{\tau}_{0:N}) p(\boldsymbol{\tau}_{0:N}) \mathrm{d}\boldsymbol{\tau}_{1:N}$$

$$= \frac{\int \xi(\boldsymbol{\tau}_{0:N}) p(\boldsymbol{\tau}_{0:N}) \mathrm{d}\boldsymbol{\tau}_{1:N}}{P(\boldsymbol{\tau}_0)} P(\boldsymbol{\tau}_0)$$

$$= \xi(\boldsymbol{\tau}_0) P(\boldsymbol{\tau}_0)$$

with $\xi(\boldsymbol{\tau}_0) = \frac{\int \xi(\boldsymbol{\tau}_{0:N}) p(\boldsymbol{\tau}_{0:N}) \mathrm{d}\boldsymbol{\tau}_{1:N}}{P(\boldsymbol{\tau}_0)}$. $\qquad \square$

# C Attaining the duality constraint on $\xi(\boldsymbol{\tau}_0)$: Proof of Proposition 4.3

*Proof of Proposition 4.3.*

*Proof of (a):* From Lemma 4.2 we know that $\xi(\boldsymbol{\tau}_0) = \frac{\int \xi(\boldsymbol{\tau}_{0:N}) p(\boldsymbol{\tau}_{0:N}) \mathrm{d}\boldsymbol{\tau}_{1:N}}{P(\boldsymbol{\tau}_0)}$. We want to have $\xi(\boldsymbol{\tau}_0) \leq \frac{1}{\alpha}$, this is equivalent to

$$\frac{\int \xi(\boldsymbol{\tau}_{0:N}) p(\boldsymbol{\tau}_{0:N}) \mathrm{d}\boldsymbol{\tau}_{1:N}}{P(\boldsymbol{\tau}_0)} \leq \frac{1}{\alpha},$$

$$\int \xi(\boldsymbol{\tau}_{0:N}) p(\boldsymbol{\tau}_{0:N}) \mathrm{d}\boldsymbol{\tau}_{1:N} \leq \frac{1}{\alpha} \int p(\boldsymbol{\tau}_{0:N}) \mathrm{d}\boldsymbol{\tau}_{1:N}.$$

One way to achieve this is to impose $\xi(\boldsymbol{\tau}_{0:N}) = \prod_{i=1}^{N} \xi(\boldsymbol{\tau}_i, \boldsymbol{\tau}_{i-1}) \leq \frac{1}{\alpha}$. This is satisfied also by constraining the single terms of the product using $\eta_i(\alpha, N)$ such that $\xi(\boldsymbol{\tau}_i, \boldsymbol{\tau}_{i-1}) \leq \eta_i(\alpha, N)$ and $\prod_{i=1}^{N} \eta_i(\alpha, N) = \frac{1}{\alpha}$.

However, $\boldsymbol{\tau}_{i-1}$ is a random quantity to which we do not have access at step $i$ of the diffusion process. Therefore, to satisfy the constraints on the single terms we impose

$$\max_{\boldsymbol{\tau}_{i-1}} \xi(\boldsymbol{\tau}_{i-1}, \boldsymbol{\tau}_i) \leq \eta_i(\alpha, N).$$

$\xi(\boldsymbol{\tau}_{i-1}, \boldsymbol{\tau}_i)$ is maximized for $(\boldsymbol{\tau}_{i-1} - \mu_{\boldsymbol{\theta}}(\boldsymbol{\tau}_i, i))^T \boldsymbol{g} < 0$, so we want $(\boldsymbol{\tau}_{i-1} - \mu_{\boldsymbol{\theta}}(\boldsymbol{\tau}_i, i))^T$ to be a vector opposite to $\boldsymbol{g}$. We can take $\boldsymbol{\tau}_{i-1} = \mu_{\boldsymbol{\theta}}(\boldsymbol{\tau}_i, i) - c_i \boldsymbol{\Sigma}_i \boldsymbol{g}_i$.

We assume that the trajectories $\boldsymbol{\tau}_i$ lie in a bounded space $C = \{\boldsymbol{\tau}_i : ||\boldsymbol{\tau}_i||_\infty \le \rho\}$, where $||\boldsymbol{\tau}_i||_\infty = \max_{\boldsymbol{s} \in \boldsymbol{\tau}_i} ||\boldsymbol{s}||_\infty$. From this assumption it follows that

$$||\boldsymbol{\tau}_{i-1}||_\infty \le \rho$$
$$||\mu_{\boldsymbol{\theta}}(\boldsymbol{\tau}_i, i) - c_i \boldsymbol{\Sigma}_i \boldsymbol{g}_i||_\infty \le \rho$$
$$||\mu_{\boldsymbol{\theta}}(\boldsymbol{\tau}_i, i)||_\infty + c_i ||\boldsymbol{\Sigma}_i \boldsymbol{g}_i||_\infty \le \rho$$
$$c_i \le \frac{(\rho - ||\mu_{\boldsymbol{\theta}}(\boldsymbol{\tau}_i, i)||_\infty)}{||\boldsymbol{\Sigma}_i \boldsymbol{g}_i||_\infty}.$$

Substituting $\boldsymbol{\tau}_{i-1} = \mu_{\boldsymbol{\theta}}(\boldsymbol{\tau}_i, i) - c_i \boldsymbol{\Sigma}_i \boldsymbol{g}_i$ into $\xi(\boldsymbol{\tau}_{i-1}, \boldsymbol{\tau}_i) \le \eta_i(\alpha, N)$ and developing we get

$$-\frac{1}{2}(-2c_i^2 \boldsymbol{g}_i^T \boldsymbol{\Sigma}_i \boldsymbol{g}_i + c_i^2 \boldsymbol{g}_i^T \boldsymbol{\Sigma}_i \boldsymbol{g}_i) \le \log \eta_i(\alpha, N)$$

$$c_i \le \sqrt{\frac{2 \log \eta_i(\alpha, N)}{\boldsymbol{g}_i^T \boldsymbol{\Sigma}_i \boldsymbol{g}_i}}.$$

So combining the two inequalities we can take

$$c_i \le \min\left( \sqrt{\frac{2 \log \eta_i(\alpha, N)}{\boldsymbol{g}_i^T \boldsymbol{\Sigma}_i \boldsymbol{g}_i}}, \frac{\rho - ||\mu_{\boldsymbol{\theta}}(\boldsymbol{\tau}_i, i)||_\infty}{||\boldsymbol{\Sigma}_i \boldsymbol{g}_i||_\infty} \right). \tag{22}$$

*Proof of (b)*: To find the minimum between the terms in Equation (22), we analyze the denominators and numerators. For the denominators, $\sqrt{\boldsymbol{g}_i^T \boldsymbol{\Sigma}_i \boldsymbol{g}_i}$ and $||\boldsymbol{\Sigma}_i \boldsymbol{g}_i||_\infty = \max_j |(\boldsymbol{\Sigma}_i \boldsymbol{g}_i)_j|$, it is equivalent to compare $\boldsymbol{g}_i^T \boldsymbol{\Sigma}_i \boldsymbol{g}_i$ and $||\boldsymbol{\Sigma}_i \boldsymbol{g}_i||_\infty^2$.

Since our diffusion model adopts a cosine scheduler for the covariance matrix, $\Sigma_i$ is diagonal with elements $(\boldsymbol{\Sigma}_i)_{jj} \in [0, 1)$. We can write the $j-$th element of $\boldsymbol{\Sigma}_i \boldsymbol{g}_i$ as $(\boldsymbol{\Sigma}_i \boldsymbol{g}_i)_j = \boldsymbol{e}_j^T \boldsymbol{\Sigma}_i \boldsymbol{g}_i$, where $\boldsymbol{e}_j$ is a basis vector. Then using Cauchy-Schwarz inequality we get

$$|(\boldsymbol{\Sigma}_i \boldsymbol{g}_i)_j|^2 = |\boldsymbol{e}_j^T \boldsymbol{\Sigma}_i \boldsymbol{g}_i|^2$$
$$\le (\boldsymbol{e}_j^T \boldsymbol{\Sigma}_i \boldsymbol{e}_j)(\boldsymbol{g}_i^T \boldsymbol{\Sigma}_i \boldsymbol{g}_i),$$

with $\boldsymbol{e}_j^T \boldsymbol{\Sigma}_i \boldsymbol{e}_j \le 1$ by definition of $\boldsymbol{\Sigma}_i$. It follows that $|(\boldsymbol{\Sigma}_i \boldsymbol{g}_i)_j|^2 \le \boldsymbol{g}_i^T \boldsymbol{\Sigma}_i \boldsymbol{g}_i$, and since this is true for all $j$ we can conclude that

$$\max_j |(\boldsymbol{\Sigma}_i \boldsymbol{g}_i)_j|^2 \le \boldsymbol{g}_i^T \boldsymbol{\Sigma}_i \boldsymbol{g}_i$$

$$||\boldsymbol{\Sigma}_i \boldsymbol{g}_i||_\infty \le \sqrt{\boldsymbol{g}_i^T \boldsymbol{\Sigma}_i \boldsymbol{g}_i}$$

When comparing the numerators of both terms in Equation (22), since $\log \eta_i(\alpha, N) = \frac{1}{N} \log\left(\frac{1}{\alpha}\right)$, for $N$ large enough, $\sqrt{2 \log \eta_i(\alpha, N)} < \rho - ||\mu_{\boldsymbol{\theta}}(\boldsymbol{\tau}_i, i)||_\infty$.

So $c_i \le \sqrt{\frac{2 \log \eta_i(\alpha, N)}{\boldsymbol{g}_i^T \boldsymbol{\Sigma}_i \boldsymbol{g}_i}}$ satisfies the dual CVaR constraints. $\qquad \square$

## D   Adaptation to Matrix Normal Distribution

Here we extend the analysis to the case where states and actions are multidimensional, and we consider trajectories as Gaussian matrices.

We define $p(\boldsymbol{\tau}_{i-1}|\boldsymbol{\tau}_i)$ as

$$p(\boldsymbol{\tau}_{i-1}|\boldsymbol{\tau}_i) = \mathcal{MN}(\boldsymbol{\tau}_{i-1}|M_{\boldsymbol{\theta}}(\boldsymbol{\tau}_i, i), \boldsymbol{U}_i, \boldsymbol{V}_i)$$

where $\mathcal{MN}(\boldsymbol{\tau}_{i-1}|M_{\boldsymbol{\theta}}(\boldsymbol{\tau}_i, i), \boldsymbol{U}_i, \boldsymbol{V}_i)$ is a Matrix Normal Distribution with mean $M_{\boldsymbol{\theta}}(\boldsymbol{\tau}_i, i) \in \mathbb{R}^{n \times p}$, row and column covariances $\boldsymbol{U}_i \in \mathbb{R}^{n \times n}$ and $\boldsymbol{V}_i \in \mathbb{R}^{p \times p}$.

The probability density function of this Matrix Normal Distribution is defined as

$$\mathcal{MN}(\boldsymbol{\tau}_{i-1}|M_{\boldsymbol{\theta}}(\boldsymbol{\tau}_i, i), \boldsymbol{U}_i, \boldsymbol{V}_i) \coloneqq K_i \exp\left(-\frac{1}{2}\text{Tr}[\boldsymbol{V}_i^{-1}(\boldsymbol{\tau}_{i-1} - M_{\boldsymbol{\theta}}(\boldsymbol{\tau}_i, i))^T \boldsymbol{U}_i^{-1}(\boldsymbol{\tau}_{i-1} - M_{\boldsymbol{\theta}}(\boldsymbol{\tau}_i, i))]\right)$$

where $\text{Tr}[\cdot]$ is the trace operator, $K_i = \frac{1}{(2\pi)^{np/2}|\boldsymbol{V}_i|^{n/2}|\boldsymbol{U}_i|^{n/2}}$ and $|\cdot|$ is the determinant of a matrix.

Define $\boldsymbol{G}_i \in \mathbb{R}^{n \times p}$ as the gradient $\nabla_{\boldsymbol{\tau}} Z$ with respect to the second order tensor representing the trajectory $\boldsymbol{\tau} \in \mathbb{R}^{n \times p}$ evaluated at $M_{\boldsymbol{\theta}}(\boldsymbol{\tau}_i, i)$. Also define $\boldsymbol{\Gamma}_i = \boldsymbol{U}_i \boldsymbol{G}_i \boldsymbol{V}_i$ for notation convenience. Consider the perturbed distribution with a mean $M_{\boldsymbol{\theta}}(\boldsymbol{\tau}_i, i) - c_i \boldsymbol{\Gamma}_i$ , we get

$$p_{\boldsymbol{\theta}}(\boldsymbol{\tau}_{i-1}|\boldsymbol{\tau}_i, \boldsymbol{\tau}_{i-1} \in C_\alpha)$$
$$= \exp\left(-\frac{1}{2}\text{Tr}[\boldsymbol{V}^{-1}(\boldsymbol{\tau}_{i-1} - M_{\boldsymbol{\theta}}(\boldsymbol{\tau}_i, i) + c_i\boldsymbol{\Gamma}_i)^T \boldsymbol{U}^{-1}(\boldsymbol{\tau}_{i-1} - M_{\boldsymbol{\theta}}(\boldsymbol{\tau}_i, i) + c_i\boldsymbol{\Gamma}_i)]\right)$$
$$= K \exp\left(-\frac{1}{2}\text{Tr}[\boldsymbol{V}^{-1}(\boldsymbol{\tau}_{i-1} - M_{\boldsymbol{\theta}}(\boldsymbol{\tau}_i, i))^T \boldsymbol{U}^{-1}(\boldsymbol{\tau}_{i-1} - M_{\boldsymbol{\theta}}(\boldsymbol{\tau}_i, i))]\right) \xi(\boldsymbol{\tau}_i, \boldsymbol{\tau}_{i-1})$$

with

$$\xi(\boldsymbol{\tau}_i, \boldsymbol{\tau}_{i-1}) = \exp\left(-\frac{1}{2}\text{Tr}[\boldsymbol{V}_i^{-1}(2c_i(\boldsymbol{\tau}_{i-1} - M_{\boldsymbol{\theta}}(\boldsymbol{\tau}_i, i))^T \boldsymbol{U}_i^{-1}\boldsymbol{\Gamma}_i + c_i^2 \boldsymbol{\Gamma}_i^T \boldsymbol{U}_i^{-1} \boldsymbol{\Gamma}_i)]\right).$$

As we did in Appendix C, we take $\xi(\boldsymbol{\tau}_i, \boldsymbol{\tau}_{i-1}) \leq \eta_i(\alpha, N)$. We can take $\boldsymbol{\tau}_{i-1} = M_{\boldsymbol{\theta}}(\boldsymbol{\tau}_i, i) - c_i \boldsymbol{\Gamma}_i$ and get

$$\exp\left(-\frac{1}{2}\text{Tr}[\boldsymbol{V}_i^{-1}(2c_i(\boldsymbol{\tau}_{i-1} - M_{\boldsymbol{\theta}}(\boldsymbol{\tau}_i, i))^T \boldsymbol{U}_i^{-1}\boldsymbol{\Gamma}_i + c_i^2 \boldsymbol{\Gamma}_i^T \boldsymbol{U}_i^{-1} \boldsymbol{\Gamma}_i)]\right) \leq \eta_i(\alpha, N)$$
$$-\frac{1}{2}\text{Tr}[\boldsymbol{V}_i^{-1}(-2c_i^2 \boldsymbol{\Gamma}_i^T \boldsymbol{U}_i^{-1}\boldsymbol{\Gamma}_i + c_i^2 \boldsymbol{\Gamma}_i^T \boldsymbol{U}_i^{-1}\boldsymbol{\Gamma}_i)] \leq \log \eta_i(\alpha, N)$$
$$\text{Tr}[\boldsymbol{V}_i^{-1}(\boldsymbol{\Gamma}_i^T \boldsymbol{U}_i^{-1}\boldsymbol{\Gamma}_i)c_i^2] \leq 2\log \eta_i(\alpha, N),$$

giving

$$c_i \leq \sqrt{\frac{2\log \eta_i(\alpha, N)}{\text{Tr}[\boldsymbol{V}_i^{-1}(\boldsymbol{U}_i\boldsymbol{G}_i\boldsymbol{V}_i)^T \boldsymbol{U}_i^{-1}(\boldsymbol{U}_i\boldsymbol{G}_i\boldsymbol{V}_i))]}}.$$

Under the same assumptions of Appendix C, we get that

$$||\boldsymbol{\tau}_{i-1}||_\infty \leq \rho$$
$$||M_{\boldsymbol{\theta}}(\boldsymbol{\tau}_i, i) - c_i\boldsymbol{U}_i\boldsymbol{G}_i\boldsymbol{V}_i||_\infty \leq \rho$$
$$||M_{\boldsymbol{\theta}}(\boldsymbol{\tau}_i, i)||_\infty + c_i||\boldsymbol{U}_i\boldsymbol{G}_i\boldsymbol{V}_i||_\infty \leq \rho$$
$$c_i \leq \frac{(\rho - ||M_{\boldsymbol{\theta}}(\boldsymbol{\tau}_i, i)||_\infty)}{||\boldsymbol{U}_i\boldsymbol{G}_i\boldsymbol{V}_i||_\infty}.$$

So we can pick

$$c_i = \min\left(\sqrt{\frac{2\log \eta_i(\alpha, N)}{\text{Tr}[\boldsymbol{V}_i^{-1}(\boldsymbol{U}_i\boldsymbol{G}_i\boldsymbol{V}_i)^T \boldsymbol{U}_i^{-1}(\boldsymbol{U}_i\boldsymbol{G}_i\boldsymbol{V}_i))]}}, \frac{(\rho - ||M_{\boldsymbol{\theta}}(\boldsymbol{\tau}_i, i)||_\infty)}{||\boldsymbol{U}_i\boldsymbol{G}_i\boldsymbol{V}_i||_\infty}\right)$$

Following the same reasoning as in Appendix C, if the covariance matrices $\boldsymbol{U}_i$ and $\boldsymbol{V}_i$ are diagonal with elements in $[0, 1)$ we can pick

$$c_i = \sqrt{\frac{2\log \eta_i(\alpha, N)}{\text{Tr}[\boldsymbol{V}_i^{-1}(\boldsymbol{U}_i\boldsymbol{G}_i\boldsymbol{V}_i)^T \boldsymbol{U}_i^{-1}(\boldsymbol{U}_i\boldsymbol{G}_i\boldsymbol{V}_i))]}}$$

# E   Implementation details

Our method makes use of three MLPs: the policy $\pi_{\boldsymbol{\omega}}$, the adversarial denoising diffusion model $\bar{p}_{\boldsymbol{\theta}}$ and the learned cumulative reward function $Z_{\phi}$.

**Policy network and training.**   The policy $\pi_{\boldsymbol{\omega}}$ is parameterized in the same way as PolyGRAD [38]. We consider a Gaussian policy of the form $\pi_{\boldsymbol{\omega}} = \mathcal{N}(\mu_{\boldsymbol{\omega}}(s), \sigma_{\boldsymbol{\omega}})$, where $\boldsymbol{\omega}$ are the parameters of the MLP. The standard deviation of the policy is a single learnable parameter $\sigma_{\boldsymbol{\omega}}$, independent of the state.

The policy is trained using Advantage Actor Critic (A2C) with Generalised Advantage Estimation (GAE). The optimizer used is ADAM. The hyperparameters can be found in Table 2.

| Parameter | Value |
|---|---|
| Batch size | 512 |
| Synthetic trajectory length | 10 |
| GAE $\lambda$ | 0.9 |
| Critic learning rate | 3e-4 |
| Actor learning rate | 3e-5 |
| Discount factor, $\gamma$ | 0.99 |
| Entropy bonus weight | 1e-5 |

Table 2: Hyperparameters for A2C training.

**Adversarial Diffusion and Cumulative Reward models.**   Our implementation builds directly on top of PolyGRAD. We follow the same training procedure, summarized in Algorithm 3. For the Diffusion Model, we use the same MLP architecture as PolyGRAD, trained by minimizing the L2 loss with ADAM optimizer. The MDP has skip connections at every layer, and features a learneable embedding of the diffusion step $i$, which is common for Diffusion Architectures [17]. The hyperparameters are summarized in Table 3.

| Parameter | Value |
|---|---|
| Hidden size | 1024 |
| Length of generated trajectory | 10 |
| Batch size | 256 |
| Diffusion step embedding size | 128 |
| Number of layers | 6 |
| Learning rate | 3e-4 |

Table 3: Hyperparameters for adversarial diffusion training.

When computing $c_i$ according to (18), we chose $\rho = 3\sigma_i$, where $\sigma_i$ is the standard deviation of the diffusion process[4] at step $i$. In our implementation we choose $\eta_i(\alpha, N) = \left(\frac{1}{\alpha}\right)^{\frac{1}{N}}$.

The cumulative reward model $Z_{\phi}$ follows the same structure and hyperparameters of the Diffusion Model (also the step embedding), with a final linear layer producing a scalar output. It is optimized using the L2 loss and the ADAM optimizer.

---

[4]In Denoising Diffusion Probabilistic Models, the standard deviation is fixed at every step $i$ according to a known scheduling rate [15].

---
**Algorithm 3** Diffusion model training
---

1: **Input:** adversarial denoising model $\bar{p}_{\boldsymbol{\theta}}$; cumulative reward function $Z_{\boldsymbol{\phi}}$; data buffer, $\mathcal{D}$; diffusion steps $N$; training iterations $K$
2: **for** $k = 1, \ldots, K$ **do**
3:      Improve $\bar{p}_{\boldsymbol{\theta}}$ (7) using $\{\boldsymbol{\tau}_0\} \sim \mathcal{D}$.
4:      Train $Z_{\boldsymbol{\phi}}$ to predict the reward $Z(\boldsymbol{\tau}_0)$.
5: **end for**

---

**Baselines implementation** The implementation of PolyGRAD was taken from the respective github repository [38]. The same was done for CPPO and M2TD3: for M2TD3, we created a new config file for Reacher, where we set the range of the actuator gear to $[50.0, 500.0]$. For TRPO and PPO we used the implementation from Stable-Baselines3 [36]. We used Domain Randomization on top of the PPO baseline, training with values of the mass and parameters uniformly sampled according to the uncertainty intervals specified in Table 4.

| Environment | Mass | Friction | Mass pole | Mass cart | Act. gear |
|---|---|---|---|---|---|
| Hopper | $[0.5, 6.5]$ | $[0.1, 3.0]$ | — | — | — |
| HalfCheetah | $[3.5, 9.5]$ | $[0.2, 0.8]$ | — | — | — |
| Walker | $[0.5, 6.5]$ | $[0.5, 2.0]$ | — | — | — |
| Cartpole | — | — | $[2.5, 10.0]$ | $[5.0, 20.0]$ | — |
| Reacher | — | — | — | — | $[50.0, 500.0]$ |

Table 4: Uncertainty sets used for domain randomization.

**Computational resources** The training of AD-RRL and Polygrad was performed on three different machines. On a cluster node with one A100 GPU, Icelake CPU and 256 GB of RAM.

The remaining model-free baselines were trained on a laptop with an Intel i7-1185G7 CPU, Mesa Intel Xe Graphics GPU and 32 GB of RAM.

In table 5 we report the wall-clock training time for each algorithm . As it is expected, the model-based algorithms (AD-RRL and PolyGRAD) are slower than the model-free ones. This is a well-known shortcoming of Model-Based RL methods, even more so when using Diffusion Models, known for their longer training times when compared to standard MLPs. AD-RRL is slower than PolyGRAD since it employes an additional Diffusion Model to approximate the cumulative reward of a trajectory.

| Algorithm | Hopper | Halfcheetah | Walker | InvertedPendulum | Reacher |
|---|---|---|---|---|---|
| AD−RRL (ours)[†] | 3-20-00 | 3-20-00 | 3-20-00 | 3-20-00 | 3-20-00 |
| Polygrad[†] | 2-14-00 | 2-14-00 | 2-14-00 | 2-14-00 | 2-14-00 |
| M2TD3[‡] | 0-02-00 | 0-03-30 | 0-02-45 | 0-02-45 | 0-02-45 |
| CPPO[‡] | 0-00-30 | 0-00-30 | 0-00-30 | 0-00-30 | 0-00-30 |
| PPO[‡] | 0-00-30 | 0-00-30 | 0-00-30 | 0-00-30 | 0-00-30 |
| TRPO[‡] | 0-00-30 | 0-00-30 | 0-00-30 | 0-00-30 | 0-00-30 |
| DR−PPO[‡] | 0-00-30 | 0-00-30 | 0-00-30 | 0-00-30 | 0-00-30 |

Table 5: Wall-clock training time (days–hours–minutes) needed to reach the reported performance on the MuJoCo tasks. Times are rounded up to the nearest quarter hour. [†]Trained on cluster node. [‡]Trained on laptop.

# F   Additional Results

In this section, we provide additional plots to support our conclusions.

## F.1   Training curves

Figure 3 shows the learning curves for AD-RRL and all the baselines for the considered MuJoCo tasks. Across seeds, AD-RRL reaches its final performance at least as quickly as the other methods. AD-RRL also achieves a final score matching or surpassing that of the baselines. This is particularly clear for the Cheetah and Walker environments, presented in Figures 3d and 3e. These results confirm that our method is more robust to modeling errors but does not sacrifice optimality in the training environment or learning speed.

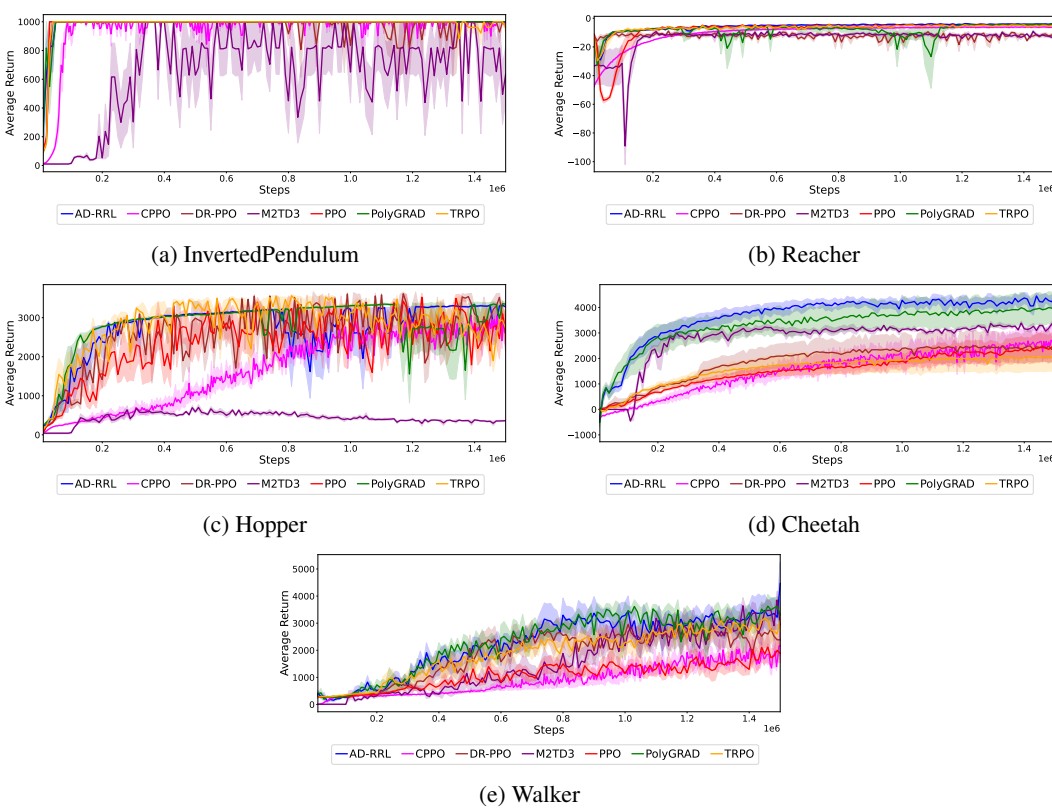

Figure 3: Training-return curves on the nominal environment for five MuJoCo tasks. Shaded areas represent one standard error over 5 runs.

## F.2 Varying parameters

In Figure 4 we present additional parameters variations for the InvertedPendulum and Cartpole environment. The pattern is consistent with the plots presented in Figure 2: AD-RRL achieves on par or higher returns than both robust and non-robust baselines as the dynamics deviate from nominal values. The only exception appears to be for higher variations of the cart mass, in the Inverted Pendulum environment (Figure 4b), where the additional inertia pushes most methods toward failure and AD-RRL similarly shows a performance decline.

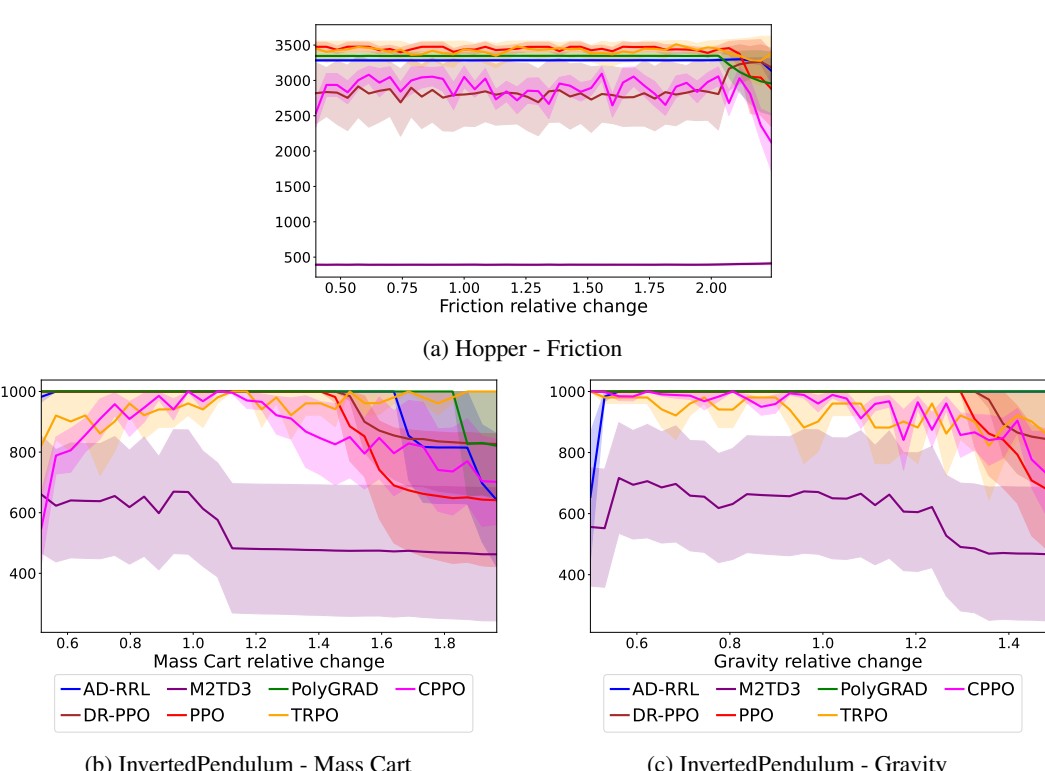

(a) Hopper - Friction

(b) InvertedPendulum - Mass Cart
(c) InvertedPendulum - Gravity

Figure 4: Average Return for varying physical parameters. Shaded areas represent one standard error over 5 runs.

## F.3 Ablation study

In this section, we compare AD-RRL trained on 1.5M environment samples with model-free baselines trained on 3M samples. This setup allows model-free methods to compensate for their lower sample efficiency by leveraging their greater computational efficiency. In Figure 5 we present the learning curves on the nominal environment for the Model-Free algorithms. We can observe that AD-RRL still outperforms or (in the worst case) matches the performance of the Model-Free baselines in all of the environments. This is also reflected when testing the robustness of the algorithms, as shown in Figure 6. AD-RRL still achieves on par or higher returns than robust and non-robust Model-Free baselines. When comparing with Figure 2, we can see that some baselines perform worse than their 1.5M samples versions (e.g., DR-PPO on the Walker and Hopper environments; TRPO on Hopper). This is expected, since an increased amount of samples might improve the performance on the nominal training environment, but will also lead to overfitting, making the agents more vulnerable to parameter changes in the test environment. The only exception appears to be M2TD3, which is able to outperform AD-RRL on the Walker environment at test time, when varying the physics parameters, despite achieving the same performance in the nominal environment. However, M2TD3 seems to be more prone to overfitting in easier environments, as seen in Figure 5a, where the baseline cannot reach the optimal solution but instead suffers a substantial performance decline in the latest training stages. Finally, M2TD3 remains stuck to a suboptimal policy in the Hopper environment, similarly to what we observed when training it on 1.5M samples. The doubled amount of samples does not seem

to benefit the learning process of this baseline. Overall, even when model-free baselines are given twice as many samples, AD-RRL remains on average better or on par across the proposed tasks.

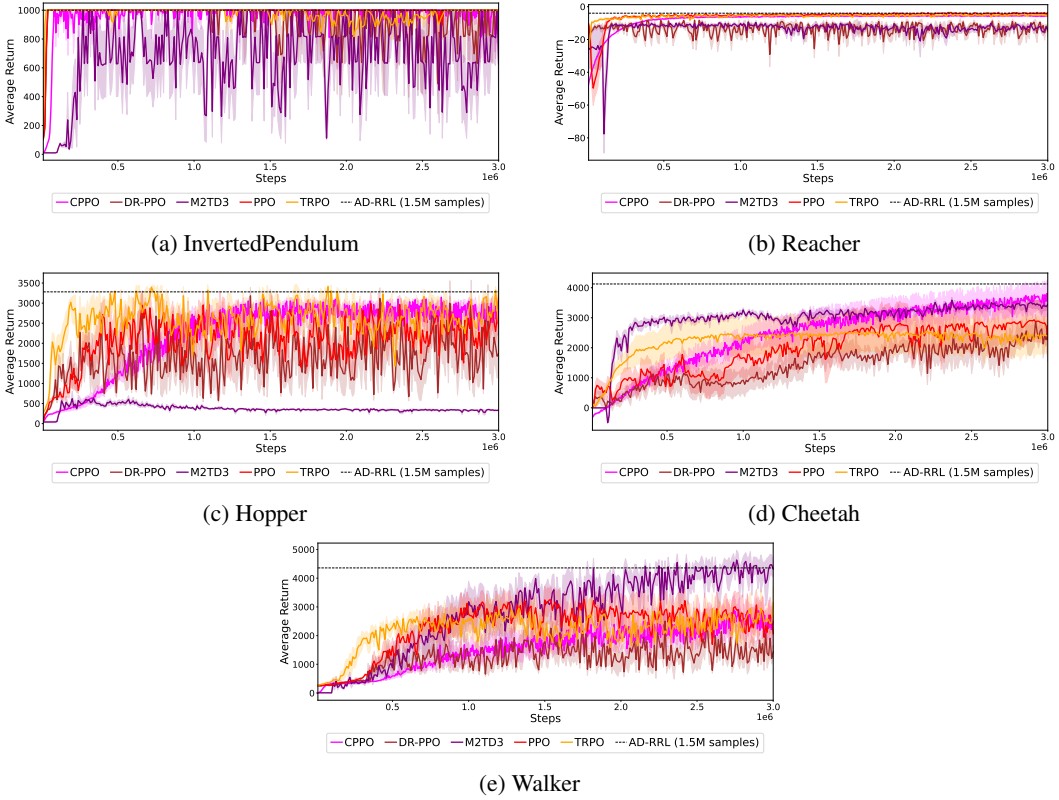

(a) InvertedPendulum

(b) Reacher

(c) Hopper

(d) Cheetah

(e) Walker

Figure 5: Training-return curves on the nominal environment for the Model-Free baslines trained on 3M samples. Shaded areas represent one standard error over 5 runs. The dashed line represents the reference final cumulative reward achieved by AD-RRL trained on 1.5M samples.

# G    Limitations and future work

**Computation time.** Guided diffusion requires dozens of reverse–diffusion steps for every synthetic trajectory and an extra gradient evaluation at each step. Consequently AD–RRL is slower than its precursor PolyGRAD, and both model–based methods need a higher training time (wall-clock) than the model-free baselines, even though they are more sample efficient. Improving the training time for diffusion models (e.g., fine-tuning the network size or the number of denoising steps) sounds like a natural next step.

**Smooth–dynamics assumption.** Our derivation employs a Gaussian approximation and the computation of gradients $\nabla_\tau Z(\tau)$. Both of these presuppose reasonably smooth rewards and state transitions. Some tasks might break this assumption, causing inaccurate guidance. Extending adversarial diffusion to domains with non-smooth dynamics is left for future work.

**Scope of the evaluation.** Our evaluation focused on simulated control tasks. A natural next step is a Sim2Real study. That is, AD–RRL is trained entirely in simulation and then deployed on real hardware, measuring how much the adversarial-diffusion training reduces the Sim2Real performance drop.

# H    Societal impact statement

Our contribution is methodological: we propose a technique for making model-based RL more robust to dynamics misspecification. Robustness is typically beneficial—e.g., safer robot control or fewer

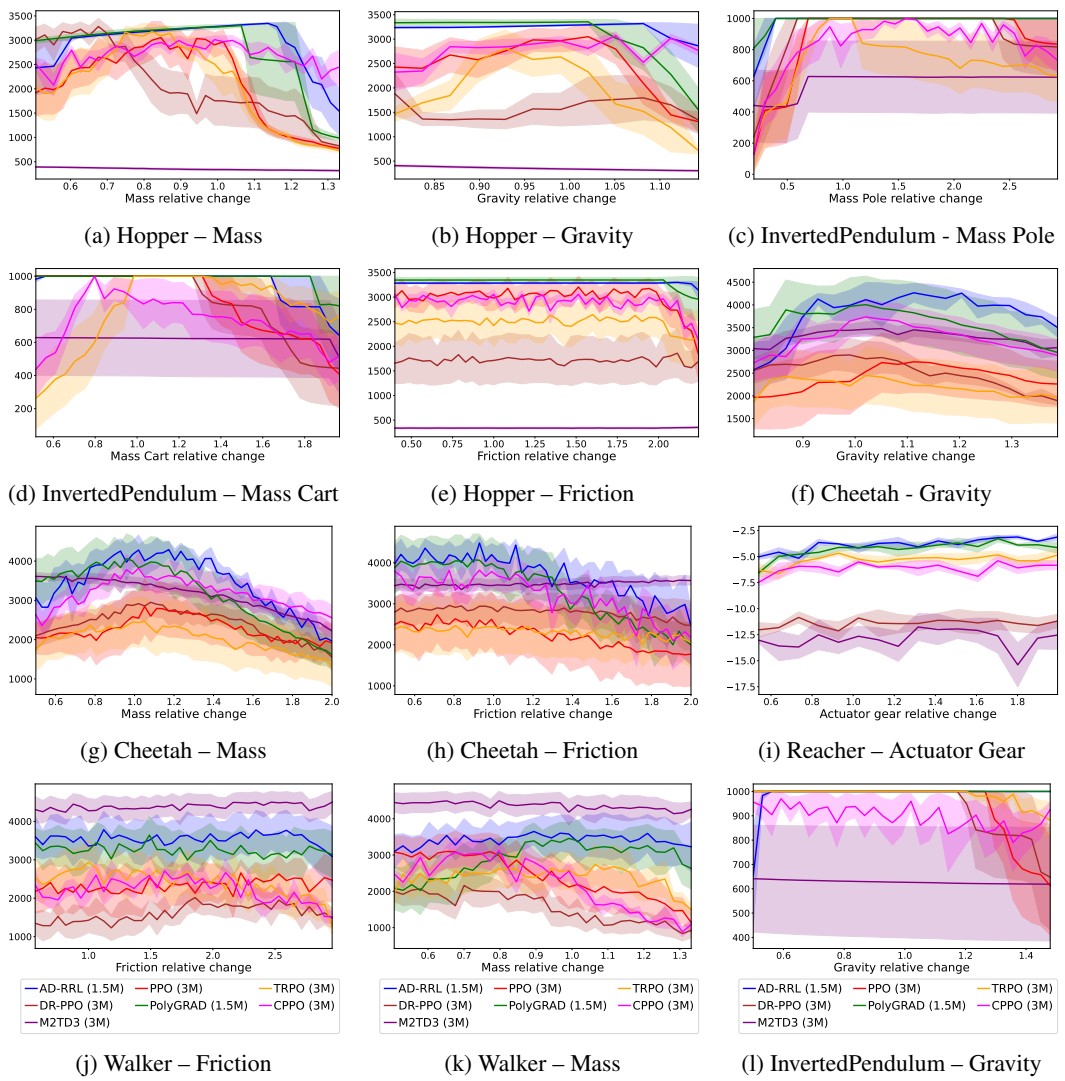

Figure 6: Average return across variations in selected physics parameters. Shaded regions indicate $\pm$ one standard error over 5 runs. AD-RRL and PolyGRAD are trained on $1.5$M samples, while the remaining Model-Free Baselines are trained on 3M samples.

failures in medical-decision support—yet any improvement in sample efficiency or policy quality can also lower the barrier to deploying RL in high-stakes settings. In domains such as healthcare, finance, or autonomous driving, deployment must therefore be accompanied by domain-specific safety checks, bias audits, and human oversight. Our work does not introduce new data-collection practices, nor does it touch sensitive attributes, but it *could* be combined with decision pipelines that do. We encourage future users of this method to evaluate downstream ethical, legal, and safety implications before real-world deployment.

