# OpenReview forum: "Adversarial Diffusion for Robust Reinforcement Learning"
_NeurIPS.cc/2025/Conference — NeurIPS 2025 poster_

### Official Review · Reviewer_TZ4u · 2025-06-17

**Clarity:** 2
**Significance:** 2
**Originality:** 3
**Rating:** 4
**Confidence:** 4

**Summary:**

This paper proposes a new solution to robust reinforcement learning under environmental uncertainty. The objective is to learn a policy that maximizes the Conditional Value-at-Risk (CVaR) of trajectory returns—a problem previously explored in the literature. The authors build on PolyGRAD, a model-based reinforcement learning (RL) method that leverages a diffusion world model and policy-guided trajectory diffusion. The main contribution of the paper is the integration of adversarial guidance into PolyGRAD to generate the worst α-percentile trajectories under a given policy, which are then used to iteratively improve policy robustness. Experimental results show that the proposed method, AD-RRL, improves both the robustness and performance of policies across several MuJoCo tasks, outperforming PolyGRAD and other baseline approaches.

**Questions:**

1. The proposed AD-RRL method is much slower than the PPO and TRPO baselines. How can it be improved using a more efficient diffusion model?
2. How does AD-RRL compare with CVaR-PPO in terms of robustness and computational efficiency?
3. The proposed adversarial diffusion guidance is based on a smooth approximation of the classifier. How tight is this approximation? What are the limitations of this approach?

**Ethical Concerns:**

["NO or VERY MINOR ethics concerns only"]

**Final Justification:**

I've reviewed the rebuttal and the comments from the other reviewers. I acknowledge that the paper has merit and appreciate the authors' efforts in addressing my concerns, so I've increased my rating from 3 to 4. However, I still think it is crucial to provide a more systematic and fair evaluation of all the baselines in the revised paper. Also, while developing a new diffusion model may be beyond the scope of this paper, it is both reasonable and feasible to adapt the approach to other existing diffusion models, such as EDM and latent diffusion models, that are known to be more efficient.

**Limitations:**

yes

**Quality:**

3

**Strengths And Weaknesses:**

The proposed method demonstrates that a diffusion world model can be adversarially guided to incorporate robustness with theoretical guarantees, highlighting the potential of diffusion models in robust sequential decision-making.

However, a major limitation of AD-RRL lies in its substantial computational demands. Trajectory diffusion is inherently slow, and the proposed adversarial guidance further increases computational complexity by requiring gradient evaluations with respect to entire trajectories. This makes the approach unlikely to scale effectively to environments with high-dimensional state spaces, such as Atari games with image-based inputs.

As shown in Table 5, training AD-RRL policies is over 100 times slower than training PPO or TRPO policies in the same environments, with even simple MuJoCo tasks taking more than three days to complete. This raises serious concerns about the practicality of the approach for real-world applications. While AD-RRL employs the same diffusion architecture as PolyGRAD, more efficient alternatives—such as latent diffusion models and EDM—have been recently proposed. To achieve a more practical solution, it is crucial to explore these more scalable methods and adapt them to trajectory diffusion.

In addition, the paper primarily compares AD-RRL to non-robust reinforcement learning baselines. However, robust RL has received increasing attention, including methods that optimize for CVaR objectives, such as CVaR-Proximal Policy Optimization (CPPO) [1]. It is expected that CPPO and similar methods incur significantly lower computational overhead compared with AD-RRL. A more meaningful comparison would include such robust RL baselines.

Table 1 shows that AD-RRL outperforms PolyGRAD, PPO, and other baselines across several MuJoCo tasks, even in the training (non-adversarial) environments. This is somewhat surprising, as robust methods like AD-RRL are typically expected to trade off performance in nominal settings for improved robustness. Further explanation is needed here.

Finally, the proposed adversarial diffusion guidance is based on a smooth approximation of the classifier. Although Proposition 4.3 shows that constants c_i can be chosen to ensure that the perturbed trajectories lie within the risk envelope, the tightness of this approximation is unclear. Including ablation studies or empirical analyses of the approximation's impact would strengthen the paper.

[1] Chengyang Ying, Towards Safe Reinforcement Learning via Constraining Conditional Value at Risk, 2021.

---

> ### Author Rebuttal · Authors · 2025-07-31
>
> We thank the reviewer for their detailed and thoughtful evaluation of our work. We are pleased that the reviewer recognized the novel integration of adversarial guidance into diffusion-based planning for robust reinforcement learning and appreciated our method’s ability to improve both robustness and performance across multiple tasks. We are also grateful for the reviewer’s recognition of the theoretical grounding of our approach and the broader potential of diffusion models in robust sequential decision-making. At the same time, we acknowledge and value the reviewer’s detailed comments, and we now provide responses to each point below.
>
> ---
>
> **Q1. The proposed AD-RRL method is much slower than the PPO and TRPO baselines. How can it be improved using a more efficient diffusion model?**
>
> Thank you for raising this important practical point. Our method is approximately $1.5\times$ more computationally expensive than PolyGRAD, which does not include adversarial guidance (note that AD-RRL requires an additional model).
>
> - As a model-based method, higher computational cost compared to model-free baselines is expected. However, we believe this overhead is justified by the significant gains in robustness and nominal performance, as demonstrated in Figures 1–3 and Table 1. AD-RRL consistently outperforms all baselines — including model-free and model-based approaches — in both nominal and perturbed environments, illustrating a favorable trade-off between computational resources and robustness.
> - Importantly, the proposed adversarial conditioning method is architecture-agnostic and can be applied on top of more efficient diffusion models. This means that AD-RRL can directly benefit from advances in efficient generative modeling, such as latent diffusion or EDM (as mentioned by the reviewer).
>
> We agree that reducing computational overhead is an important avenue for future research and consider integrating such efficient architectures a promising direction.
>
> ---
>
> **Q2. How does AD-RRL compare with CVaR-PPO in terms of robustness and computational efficiency?**
>
> We thank the reviewer for suggesting this additional baseline. For a fair comparison, we trained CVaR-PPO (CPPO) for 1.5 M steps — consistent with AD-RRL and the other baselines — and present point-wise evaluations below.
>
> |  | HalfCheetah |  |
> | --- | --- | --- |
> | **Mass relative change** | **AD-RRL (Ours)** | **CPPO** |
> | 0.500 | **3061.153 ± 548.862** | −73.265 ± 120.968 |
> | 0.806 | **3898.686 ± 414.957** | 1555.600 ± 351.794 |
> | 1.112 | **4198.342 ± 388.928** | 2488.860 ± 453.486 |
> | 1.265 | **3716.105 ± 282.915** | 1823.855 ± 370.183 |
> | 1.725 | **2691.277 ± 279.085** | 1297.462 ± 344.145 |
>
> *Table 1 — Performance across mass-relative changes for HalfCheetah (best results in bold).*
>
>
> |  | Hopper |  |
> | --- | --- | --- |
> | **Mass relative change** | **AD-RRL (Ours)** | **CPPO** |
> | 0.506 | **2431.643 ± 543.152** | **2035.358 ± 350.707** |
> | 0.627 | **3062.959 ± 10.649** | 2617.749 ± 204.166 |
> | 0.749 | **3147.467 ± 12.367** | 2902.990 ± 77.162 |
> | 0.870 | **3215.967 ± 15.982** | 2603.035 ± 215.011 |
> | 1.161 | **3280.748 ± 81.662** | 2189.888 ± 379.293 |
>
> *Table 2 — Performance across mass-relative changes for Hopper (best results in bold).*
>
> |  | Walker |  |
> | --- | --- | --- |
> | **Mass relative change** | **AD-RRL (Ours)** | **CPPO** |
> | 0.506 | **3095.562 ± 467.163** | 1045.981 ± 171.053 |
> | 0.627 | **3293.219 ± 368.472** | 1366.939 ± 131.585 |
> | 0.749 | **3364.389 ± 363.136** | 1514.123 ± 124.977 |
> | 0.870 | **3556.677 ± 384.626** | 2033.778 ± 86.922 |
> | 1.161 | **3361.651 ± 644.848** | 831.669 ± 55.307 |
>
> *Table 3 — Performance across mass-relative changes for Walker (best results in bold).*
>
>
> In terms of efficiency, one CPPO run takes roughly 30 minutes, compared to slightly under 4 days for AD-RRL. This is expected, since CPPO is a model-free method that does not rely on learned dynamics or trajectory guidance. However, as the tables show, AD-RRL vastly outperforms CPPO and is significantly more robust to variations in environment parameters. Its performance slowly degrades as the mass parameter shifts, whereas CPPO’s performance declines sharply. We believe this greatly increased robustness justifies the additional overhead, and again highlight the future potential of incorporating more efficient architectures into AD-RRL.
>
> ---
>
> **Q3. The proposed adversarial diffusion guidance is based on a smooth approximation of the classifier. How tight is this approximation? What are the limitations of this approach?**
>
> Thank you for this interesting question.
>
> Our approach builds on prior work (e.g., [1]), which casts control as probabilistic inference. There, the probability of optimality at each time step is approximated as $P(O_t = 1 \mid s_t, a_t) = \exp(r(s_t, a_t)),$ a smooth relaxation of the hard indicator $P(O_t = 1 \mid s_t, a_t) = \mathbf{1}_{(s_t, a_t) \in C}$.
>
> - In our setting, a diffusion model generates entire trajectories rather than individual actions. We therefore model the optimality of an entire trajectory $\tau$:
>   - Let $O(\tau)$ be a binary variable indicating whether a trajectory is optimal. We approximate $P \bigl(O(\tau)=1 \mid \tau\bigr) = \exp\bigl(-c_0 \sum_t \gamma^t r_t\bigr),$ where $c_0$ controls the sharpness of the approximation.
>   - This smooth form allows gradient-based guidance to steer the diffusion model toward low-return (i.e., adversarial) trajectories.
>   - Without such an approximation, adversarial guidance would be intractable in diffusion-based planning.
>
> - The tightness of this approximation is governed by the scalar $c_0$:
>   - Larger values of $c_0$ make the approximation closer to a hard threshold; smaller values produce smoother weighting.
>   - Similarly, at each diffusion step $i$, a corresponding $c_i$ governs the sharpness of the step-specific approximation.
>
> - In practice, we observe that $c_i$ increases as $i$ decreases (i.e., as the model approaches the final, denoised trajectory $\tau_0$), making the approximation sharper at later steps. This effect arises from Eq. (19) and from the cosine noise schedule of the diffusion model [2], which reduces variance as $i$ decreases — thus increasing $c_i$.
>
> - A limitation is that this smooth approximation may not sharply distinguish trajectories that lie near the CVaR threshold. Nevertheless, it provides a practical and differentiable surrogate for otherwise intractable objectives.
>
> We acknowledge that exploring tighter or alternative approximations, and empirically validating their effects, would be a valuable direction for future work and will mention this in the final version.
>
> ---
>
> **Q4. AD-RRL outperforms PolyGRAD, PPO, and other baselines across several MuJoCo tasks, even in the training (non-adversarial) environments.**
>
> Indeed, while AD-RRL is designed for robustness, it also performs strongly in the nominal setting. We believe this stems from two main factors:
>
> - First, the CVaR objective lower-bounds the expected return. Maximizing CVaR thus implicitly promotes strong average performance.
>
> - Second, adversarial guidance encourages the policy to learn from failure modes that might be underexplored in other approaches. This leads to more informative training signals and improved generalization, even under nominal dynamics.
>
> --
>
> We hope that, given the detailed comparison with CVaR-PPO and our in-depth answers to the reviewer's questions, they might consider raising the final score for our paper.
>
> --
>
> [1] Levine, Sergey. "Reinforcement learning and control as probabilistic inference: Tutorial and review." arXiv preprint arXiv:1805.00909 (2018).
>
> [2] Nichol, Alexander Quinn, and Prafulla Dhariwal. "Improved denoising diffusion probabilistic models." International conference on machine learning. PMLR, 2021.

---

> > ### Comment · Reviewer_TZ4u · 2025-08-02
> >
> > I appreciate the authors' clarification and additional results. However, I'm still unconvinced that an RL algorithm that uses 4 days to solve simple MuJoCo tasks is practical. The fact that the algorithm is 1.5 times more computationally expensive than PolyGRAD, which this paper heavily relies upon, does not fully justify the added complexity on its own.
> >
> > The new results are also not particularly convincing. The CPPO paper [1] demonstrates that their algorithm continues to improve after 3 million steps, roughly equivalent to one hour of training. Capping training at 1.5 million steps, therefore, appears arbitrary and difficult to justify. This concern applies to all results reported in the paper, where most baselines are less expensive than AD-RRL, yet are uniformly capped at 1.5 million steps. This choice seems unfair to those baselines, which, like CPPO, would likely continue improving if allowed to train longer. The lack of cumulative reward curves over training steps further limits the ability to evaluate whether the baselines reach their full potential.
> >
> > [1] Chengyang Ying, Towards Safe Reinforcement Learning via Constraining Conditional Value at Risk, 2021.

---

> > > ### Author Response · Authors · 2025-08-03
> > > **Clarifications on computational complexity and new additional results**
> > >
> > > We thank the reviewer for their thoughtful follow-up and the opportunity to further clarify our contributions. We appreciate the careful consideration of our results and the feedback regarding runtime, baseline comparisons, and learning curves. We address each point in turn below.
> > >
> > > ---
> > >
> > > ### **Q1. Still unconvinced that the method is practical. The fact that the algorithm is 1.5 times more computationally expensive than PolyGRAD does not fully justify the added complexity on its own.**
> > >
> > > As mentioned in our initial reply, AD-RRL adds an additional model (trained jointly with the adversarial guidance module) on top of PolyGRAD, which increases training time by roughly $1.5\times$. This additional component is used to predict the cumulative reward of the generated trajectories, for every diffusion step. Therefore, it requires additional training, and adds computational complexity. For more details, we refer the reviewer to Section&nbsp;5 and Appendix&nbsp;E of our paper.
> > >
> > > It is a bit shortsighted to disregard a method only for being more computationally intensive. It is beyond the scope of the paper to propose a more efficient diffusion model, but we believe that, given the speed at which the field is moving, this issue may become less relevant in the future.
> > >
> > > Finally, we respectfully remind the reviewer that many now-standard methods (e.g., DQN, TRPO, Dreamer) were initially considered computationally demanding. Over time, more efficient architectures and training tricks made them practical. We see AD-RRL as a principled foundation that can be improved by future work — for example, by replacing its diffusion backbone with more efficient generative models.
> > >
> > > ---
> > >
> > > ### **Q2. Capping training at 1.5 million steps appears arbitrary and unfair to faster baselines like CPPO. This concern applies to all results reported in the paper, where most baselines are less expensive than AD-RRL, yet are uniformly capped at 1.5 million steps.**
> > >
> > > While we understand the concern of the reviewer, we kindly disagree that the choice is unjustified. Indeed, we have the following two facts:
> > >
> > > 1. The 1.5M step cap follows a common practice in recent RL literature (e.g., [1,2,3,4]), where many algorithms converge within 1–2 M steps.
> > > 2. In Appendix F we report the training curves for all baselines, showing that the majority of methods — including PPO, TRPO, DR-PPO, and PolyGRAD — plateau well before 1.5 M steps.
> > >
> > > However, we followed the reviewer's suggestion and further trained CPPO for 6 M steps. AD-RRL is still trained for 1.5 M steps. We present the results in the tables in our next comment (due to character limits).
> > >
> > > While CPPO performance improves, AD-RRL maintains stronger robustness and nominal performance. We would like to point out that AD-RRL always outperforms CPPO in average performance. We highlight in **bold** the best-performing algorithm for each mass variation: in a few cases, the values overlap when adding the standard error, so we highlight both for a fairer comparison. We also note that even in the CPPO paper itself, some baselines (e.g., TRPO) had not yet converged at 3 M steps. This underscores a general challenge in RL: convergence speed varies widely across methods.
> > >
> > > We agree that different methods may benefit from different budgets, but we chose a fixed budget for fairness and reproducibility — a standard approach in benchmark evaluations.
> > >
> > > ---
> > >
> > > ### **Q3. The lack of cumulative reward curves over training steps limits the ability to evaluate convergence.**
> > >
> > > Thank you for highlighting this. As mentioned above, we do include training curves in Appendix F for all baseline methods. These curves show that nearly all algorithms reach convergence by 1.5 M steps.
> > >
> > > Unfortunately, given the restrictions imposed by NeurIPS this year, we are not able to attach additional plots presenting the learning curve for CPPO. However, we are happy to share the evaluation points for the last 100 k steps. We present the results in the tables in our next comment (due to character limits). These show that CPPO performance does plateau by 6 M steps. We hope the comparison in performance with AD-RRL will now look fair.
> > >
> > > If the remaining doubt for the reviewer is the computational efficiency of AD-RRL, we are confident that our method will be improved by future research. If the reviewer would like to see more results that could help them lean towards acceptance we will do our best to deliver them by the August 6th deadline. Otherwise, since the previous questions have been answered in great detail, we hope the reviewer will consider raising their final grade.
> > >
> > > ---
> > >
> > > [1] Scott Fujimoto, et al. Addressing function approximation error in actor-critic methods. ICML 2018.
> > >
> > > [2] Yasuhiro Fujita, et al. Clipped action policy gradient. ICML 2018.
> > >
> > > [3] Changling Li, et al. Regularized optimal experience replay. arXiv 2024.
> > >
> > > [4] Susan Amin, et al. Locally persistent exploration in continuous control tasks with sparse rewards. arXiv, 2020.

---

> > > > ### Author Response · Authors · 2025-08-03
> > > > **Tables for our previous comment**
> > > >
> > > > ## **Performance under mass changes on HalfCheetah**
> > > >
> > > > | Mass relative change | AD-RRL (Ours) | CPPO (6 M steps) |
> > > > | --- | --- | --- |
> > > > | 0.500 | $\boldsymbol{3061.153 \pm 548.862}$ | $428.598 \pm 840.775$ |
> > > > | 0.806 | $\boldsymbol{3898.686 \pm 414.957}$ | $2460.237 \pm 1325.258$ |
> > > > | 1.112 | $\boldsymbol{4198.342 \pm 388.928}$ | $\boldsymbol{3438.553 \pm 1204.456}$ |
> > > > | 1.265 | $\boldsymbol{3716.105 \pm 282.915}$ | $2153.036 \pm 945.458$ |
> > > > | 1.725 | $\boldsymbol{2691.277 \pm 279.085}$ | $1262.486 \pm 864.017$ |
> > > >
> > > > ---
> > > >
> > > > ## **Performance under mass changes on Hopper**
> > > >
> > > > | Mass relative change | AD-RRL (Ours) | CPPO (6 M steps) |
> > > > | --- | --- | --- |
> > > > | 0.506 | $\boldsymbol{2431.643 \pm 543.152}$ | $\boldsymbol{2201.615 \pm 977.578}$ |
> > > > | 0.627 | $\boldsymbol{3062.959 \pm 10.649}$ | $2123.093 \pm 950.427$ |
> > > > | 0.749 | $\boldsymbol{3147.467 \pm 12.367}$ | $2893.158 \pm 234.081$ |
> > > > | 0.870 | $\boldsymbol{3215.967 \pm 15.982}$ | $\boldsymbol{2786.064 \pm 688.433}$ |
> > > > | 1.161 | $\boldsymbol{3280.748 \pm 81.662}$ | $2295.719 \pm 695.671$ |
> > > >
> > > > ---
> > > >
> > > > ## **Performance under mass changes on Walker**
> > > >
> > > > | Mass relative change | AD-RRL (Ours) | CPPO (6 M steps) |
> > > > | --- | --- | --- |
> > > > | 0.506 | $\boldsymbol{3095.562 \pm 467.163}$ | $1142.072 \pm 304.613$ |
> > > > | 0.627 | $\boldsymbol{3293.219 \pm 368.472}$ | $1593.641 \pm 456.268$ |
> > > > | 0.749 | $\boldsymbol{3364.389 \pm 363.136}$ | $2823.858 \pm 391.522$ |
> > > > | 0.870 | $\boldsymbol{3556.677 \pm 384.626}$ | $2963.500 \pm 452.328$ |
> > > > | 1.161 | $\boldsymbol{3361.651 \pm 644.848}$ | $877.010 \pm 249.669$ |
> > > >
> > > > ---
> > > >
> > > > ## **HalfCheetah training snapshots for CPPO**
> > > >
> > > > | Training steps | Average return | Standard Error |
> > > > | --- | --- | --- |
> > > > | 5 900 000 | 4 058.55 | 421.40 |
> > > > | 5 904 000 | 4 311.52 | 583.87 |
> > > > | 5 908 000 | 3 860.28 | 403.98 |
> > > > | 5 912 000 | 3 643.88 | 457.98 |
> > > > | 5 916 000 | 4 008.06 | 589.06 |
> > > > | 5 920 000 | 4 065.86 | 528.96 |
> > > > | 5 924 000 | 3 854.60 | 416.94 |
> > > > | 5 928 000 | 3 959.99 | 412.50 |
> > > > | 5 932 000 | 4 286.19 | 585.08 |
> > > > | 5 936 000 | 3 979.35 | 444.94 |
> > > > | 5 940 000 | 4 308.76 | 562.70 |
> > > > | 5 944 000 | 4 144.92 | 487.86 |
> > > > | 5 948 000 | 4 194.36 | 575.84 |
> > > > | 5 952 000 | 4 300.33 | 577.89 |
> > > > | 5 956 000 | 3 937.57 | 414.54 |
> > > > | 5 960 000 | 4 340.71 | 575.22 |
> > > > | 5 964 000 | 4 231.08 | 591.15 |
> > > > | 5 968 000 | 4 360.17 | 587.72 |
> > > > | 5 972 000 | 4 098.65 | 601.60 |
> > > > | 5 976 000 | 4 214.84 | 554.52 |
> > > > | 5 980 000 | 3 896.63 | 360.42 |
> > > > | 5 984 000 | 4 216.35 | 611.58 |
> > > > | 5 988 000 | 3 999.67 | 418.90 |
> > > > | 5 992 000 | 4 142.91 | 658.65 |
> > > > | 5 996 000 | 4 234.14 | 615.40 |
> > > > | 6 000 000 | 3 886.47 | 352.66 |
> > > >
> > > > ---
> > > >
> > > > ## **Hopper training snapshots for CPPO**
> > > >
> > > > | Training steps | Average return | Standard Error |
> > > > | --- | --- | --- |
> > > > | 5 900 000 | 2 578.78 | 150.56 |
> > > > | 5 904 000 | 2 963.40 | 162.25 |
> > > > | 5 908 000 | 2 922.37 | 86.15 |
> > > > | 5 912 000 | 2 831.37 | 224.17 |
> > > > | 5 916 000 | 3 099.87 | 55.47 |
> > > > | 5 920 000 | 3 020.14 | 82.96 |
> > > > | 5 924 000 | 3 133.21 | 58.24 |
> > > > | 5 928 000 | 3 121.01 | 62.34 |
> > > > | 5 932 000 | 2 638.40 | 295.55 |
> > > > | 5 936 000 | 2 922.15 | 205.56 |
> > > > | 5 940 000 | 3 121.29 | 58.07 |
> > > > | 5 944 000 | 3 130.47 | 51.22 |
> > > > | 5 948 000 | 3 007.93 | 148.01 |
> > > > | 5 952 000 | 3 040.14 | 113.47 |
> > > > | 5 956 000 | 3 048.40 | 77.15 |
> > > > | 5 960 000 | 3 140.67 | 42.35 |
> > > > | 5 964 000 | 3 077.03 | 87.75 |
> > > > | 5 968 000 | 3 009.03 | 147.20 |
> > > > | 5 972 000 | 3 162.71 | 24.43 |
> > > > | 5 976 000 | 3 126.45 | 40.90 |
> > > > | 5 980 000 | 3 153.54 | 27.43 |
> > > > | 5 984 000 | 3 105.03 | 96.46 |
> > > > | 5 988 000 | 2 845.27 | 159.44 |
> > > > | 5 992 000 | 3 017.71 | 185.28 |
> > > > | 5 996 000 | 3 012.10 | 206.70 |
> > > > | 6 000 000 | 2 717.23 | 216.06 |
> > > >
> > > > ---
> > > >
> > > > ## **Walker training snapshots for CPPO**
> > > >
> > > > | Training steps | Average return | Standard Error |
> > > > | --- | --- | --- |
> > > > | 5 900 000 | 3 070.34 | 56.13 |
> > > > | 5 904 000 | 2 675.65 | 288.12 |
> > > > | 5 908 000 | 2 789.79 | 144.53 |
> > > > | 5 912 000 | 2 637.37 | 217.44 |
> > > > | 5 916 000 | 2 938.86 | 193.75 |
> > > > | 5 920 000 | 2 882.20 | 241.37 |
> > > > | 5 924 000 | 3 157.27 | 258.31 |
> > > > | 5 928 000 | 3 020.40 | 370.93 |
> > > > | 5 932 000 | 2 654.32 | 235.51 |
> > > > | 5 936 000 | 3 415.86 | 376.42 |
> > > > | 5 940 000 | 2 775.72 | 195.18 |
> > > > | 5 944 000 | 2 660.08 | 130.83 |
> > > > | 5 948 000 | 2 659.32 | 280.03 |
> > > > | 5 952 000 | 3 034.52 | 263.01 |
> > > > | 5 956 000 | 3 082.96 | 419.15 |
> > > > | 5 960 000 | 3 288.25 | 310.69 |
> > > > | 5 964 000 | 2 895.52 | 272.06 |
> > > > | 5 968 000 | 3 492.83 | 184.83 |
> > > > | 5 972 000 | 2 881.44 | 379.63 |
> > > > | 5 976 000 | 2 734.65 | 234.19 |
> > > > | 5 980 000 | 2 853.65 | 387.42 |
> > > > | 5 984 000 | 2 477.36 | 293.51 |
> > > > | 5 988 000 | 2 835.29 | 260.16 |
> > > > | 5 992 000 | 2 744.84 | 152.13 |
> > > > | 5 996 000 | 2 656.33 | 60.79 |
> > > > | 6 000 000 | 2 709.67 | 213.92 |

---

> > > > > ### Comment · Reviewer_TZ4u · 2025-08-06
> > > > >
> > > > > I thank the authors for providing the new results, which have addressed some of my concerns. However, there are a few points in the authors' response that I respectfully disagree with.
> > > > >
> > > > > First, I do not believe it is fair to use the same number of training time steps for AD-RRL and the other baselines, given that AD-RRL is significantly more time- and resource-intensive. It is acceptable to do so in CPPO, as its computational complexity is comparable to that of the other baselines. Therefore, it is crucial to provide a more systematic evaluation of all the baselines in the revised paper.
> > > > >
> > > > > Second, AD-RRL is not a fundamentally new approach, as it heavily builds upon the trajectory diffusion framework of PolyGRAD, using the same diffusion model as PolyGRAD, differing only in the adversarial guidance part. While developing a new diffusion model may be beyond the scope of this paper, it is both reasonable and feasible to adapt the approach to other existing diffusion models, such as EDM and latent diffusion models, that are known to be more efficient.
> > > > >
> > > > > That said, I acknowledge that the paper has merit and appreciate the authors' efforts in addressing my concerns. I'll raise my score from 3 to 4.

---

> > > > > > ### Author Response · Authors · 2025-08-06
> > > > > > **Final clarifications and thanks**
> > > > > >
> > > > > > Thank you very much for the positive feedback and for reconsidering your evaluation. We will account for your comments in the final version of the manuscript.
> > > > > >
> > > > > > In the final version of the manuscript, to ensure a fair comparison, we will include additional evaluations for baselines that are more computationally efficient than AD-RRL, as that we have conducted for CPPO in this rebuttal (CPPO trained using 6M environment steps vs 1.5M steps for AD-RRL).
> > > > > >
> > > > > > As you are pointing out, our main contribution lies in introducing an adversarial guidance mechanism aligned with the CVaR framework for robust RL. We agree that adapting this framework to more efficient diffusion architectures (e.g., EDM or latent diffusion) is interesting, and we will investigate this.

---

> > > > > > > ### Author Response · Authors · 2025-08-08
> > > > > > >
> > > > > > > We would like to thank again the reviewer for the feedback and for deciding to raise their score (please do not forget to do it!).
> > > > > > >
> > > > > > > We will include the experiments that we discussed in the revised version of our paper.
> > > > > > >
> > > > > > > Many thanks!

---

### Official Review · Reviewer_2eCV · 2025-06-27

**Clarity:** 2
**Significance:** 3
**Originality:** 2
**Rating:** 4
**Confidence:** 2

**Summary:**

The paper proposes a diffusion model to generates trajectories which are biased towards low returns. These low returns trajectories are then used to train the RL models yielding robust policies.

**Questions:**

***Q1) Could  the work be compared and contrasted with  [1] that generates trajectories with worst kernel with theoretical guarantees?***

***Q2) The paper is well motivated and the approach looks good. Can we say anything about theoretical performance of the approach?***

***Q3) Could you please summarize the key challenges of the paper?***

***Q4) What are the limitations of the work?***

***Q5) There are many works on uncertainty sets bounded by $L_p$ balls, revealing many structural properties such as 'adversary is rank one perturbation of the nominal kernel' [2], etc..  Can these properties be used to design better diffusion based algorithms for robust RL?***


[2] @inproceedings{10.5555/3666122.3668721,
author = {Kumar, Navdeep and Derman, Esther and Geist, Matthieu and Levy, Kfir and Mannor, Shie},
title = {Policy gradient for rectangular robust Markov decision processes},
year = {2023},
publisher = {Curran Associates Inc.},
address = {Red Hook, NY, USA},
booktitle = {Proceedings of the 37th International Conference on Neural Information Processing Systems},
articleno = {2599},
numpages = {25},
location = {New Orleans, LA, USA},
series = {NIPS '23}
}






[1]@inproceedings{
gadot2024bring,
title={Bring Your Own (Non-Robust) Algorithm to Solve Robust {MDP}s by Estimating The Worst Kernel},
author={Uri Gadot and Kaixin Wang and Navdeep Kumar and Kfir Yehuda Levy and Shie Mannor},
booktitle={Forty-first International Conference on Machine Learning},
year={2024},
url={https://openreview.net/forum?id=UqoG0YRfQx}
}

**Ethical Concerns:**

["NO or VERY MINOR ethics concerns only"]

**Final Justification:**

Authors response is convincing, hence I am increasing the score 4 but I don't find myself very familiar with the topic, hence I reduce my confidence to 2.

**Paper Formatting Concerns:**

Seems ok.

**Quality:**

2

**Strengths And Weaknesses:**

Strengths: 1) The paper is well motivated. 2) The claims and approach seems intuitive and reasonable.

Weakness: 1) The presentation of the paper can be improved, maybe using block diagrams, illustration of the approach etc... 2) See the questions.

---

> ### Author Rebuttal · Authors · 2025-07-31
>
> We thank the reviewer for the helpful comments and valuable questions. We’re pleased to hear that the paper was found to be well-motivated, with an approach that is both intuitive and reasonable. Below, we address each comment in turn.
>
> ---
>
> **Q1. Could the work be compared and contrasted with [1] that generates trajectories with worst kernel with theoretical guarantees?**
>
> Thank you for pointing to this relevant work. The key idea of EWoK presented in [1] is to bias transitions toward next states with lower estimated value, thereby encouraging the policy to learn from pessimistic outcomes.
>
> However, there are a number of distinctions to highlight:
>
> 1. EWoK assumes access to a generative model, allowing multiple samples to be drawn from any $(s,a)$ pair. AD-RRL does not.
>    - This assumption is often impractical in real-world environments where the agent cannot reset and resample transitions on demand.
>    - In contrast, AD-RRL learns a dynamics model and uses it to generate entire trajectories into the future, enabling adversarial sampling without requiring additional environment access during training.
>
> 2. While EWoK can offer strong theoretical guarantees due to its assumption of a known transition function, AD-RRL does not make such assumptions. We believe this leads to a more realistic framework for robust RL in environments with unknown or partially observable dynamics.
>
> We agree that comparing AD-RRL with EWoK would be valuable for future work, and we will strive to include such a comparison in the final version of the manuscript.
>
> ---
>
> **Q2. The paper is well motivated and the approach looks good. Can we say anything about theoretical performance of the approach?**
>
> Thank you for the positive feedback and for this question.
>
> - In the paper, we explore how the dual CVaR formulation connects to robustness. In particular, Proposition&nbsp;4.3 formally describes how to select the scalar $c_i$ such that the adversarial guidance at each diffusion step produces trajectories within the CVaR risk envelope.
> - Choosing $c_i$ in this way ensures that the perturbations introduced by the adversarial objective remain valid from a risk-sensitive optimization perspective, and provides a principled foundation for the guided sampling process. Note that $c_i$ controls the sharpness of the approximation to the indicator function $p_\theta(\bar O(\tau_i) = 1 \mid \tau_i) = 1_{\tau_i \in C_\alpha}$ used in the CVaR formulation (see, e.g., Eq.&nbsp;12 and&nbsp;13).
>   - Larger values of $c_i$ make the function sharper (closer to a hard threshold), while smaller values smooth the weighting more broadly.
> - In practice, the values of $c_i$ increase as $i$ decreases. This means that as the model approaches the final denoised trajectory $\tau_0$, the guidance becomes more selective, making the approximation sharper and closer to a hard threshold.
>   - This trend results from the definition of $c_i$ (see Eq. 19) and the diffusion model’s cosine schedule.
>   - We use the cosine schedule proposed by [3], which decreases the variance of the diffusion model as $i$ decreases. This reduces the denominator in Eq.&nbsp;19 and thus increases $c_i$ as $i$ decreases.
>
> While we acknowledge that exploring other theoretical properties—such as regret bounds for AD-RRL—is an exciting direction, we consider this beyond the scope of the current paper.
>
> ---
>
> **Q3. Could you please summarize the key challenges of the paper?**
>
> Thank you for the question. The key challenges we address in this work are:
>
> - **Learning robust policies in uncertain environments:** Robust RL aims to prepare agents for rare or worst-case scenarios, but identifying and training on such situations is non-trivial—especially when they are not encountered during standard exploration.
> - **Generating informative, low-return trajectories:** Our challenge is to guide a diffusion model to generate *low-return (adversarial) trajectories* from the tail of the return distribution (i.e., within a CVaR risk envelope), while ensuring they remain realistic and actionable.
> - **Providing theoretical guarantees on adversarial guidance:** We derive and prove conditions (Proposition&nbsp;4.3) under which the gradient guidance coefficients $c_i$ ensure that the perturbed samples remain within the CVaR risk envelope. This offers a principled foundation for adversarial sampling throughout the diffusion process.
> - **Maintaining model fidelity while increasing robustness:** Training on adversarial samples can cause a mismatch with the data distribution. A key challenge is to ensure that the generated trajectories remain likely under the current policy, which we address by leveraging PolyGRAD’s policy-conditioned diffusion model.
>
> ---
>
> **Q4. What are the limitations of the work?**
>
> As also discussed in Appendix&nbsp;G of our paper, we acknowledge the following key limitations:
>
> - **Computational overhead:** While AD-RRL significantly improves robustness and performance, it does so at the cost of increased computational overhead, due to both the learned diffusion model and the adversarial guidance mechanism. As discussed in our response to Reviewer TZ4u, we believe this cost is justified by the observed robustness and nominal performance gains, and could be mitigated in future work by incorporating more efficient generative architectures.
> - **Smoothness assumptions:** Our derivation employs a Gaussian approximation and requires computing gradients $\nabla_{\tau}Z(\tau)$, which presupposes reasonably smooth rewards and state transitions. Tasks with non-smooth dynamics may violate this assumption, leading to less accurate guidance. Extending adversarial diffusion to such domains is an important direction for future work.
> - **Benchmark:** Finally, our experiments are currently limited to continuous control tasks in MuJoCo. While these are standard benchmarks, evaluating AD-RRL in higher-dimensional settings—such as vision-based environments (e.g., Atari)—would further clarify the method’s scalability.
>
> ---
>
> **Q5. There are many works on uncertainty sets bounded by $L_p$ balls, revealing many structural properties such as “adversary is rank-one perturbation of the nominal kernel” [2], etc. Can these properties be used to design better diffusion-based algorithms for robust RL?**
>
> Thank you for the insightful question. Indeed, many works in robust MDPs (e.g., [2]) define uncertainty sets using $L_p$-norm balls around nominal transition kernels, which leads to useful structural properties such as rectangularity, convexity, and tractable dual forms.
>
> Our method generates samples at the trajectory level using a learned diffusion model, which makes it difficult to directly apply kernel-level uncertainty set formulations.
>
> That said, these structural insights could certainly inspire future improvements. For instance, incorporating constraints or priors on the learned dynamics model (e.g., Lipschitz continuity or rectangularity) may lead to better-calibrated adversarial sampling and stronger theoretical guarantees.
>
> ---
>
> **Q6. Clarity of presentation**
>
> Thank you for suggesting a concrete way to improve the clarity of our paper. In the final version, we will include an additional illustration to provide a simplified overview of the AD-RRL pipeline.
>
> --
>
> We hope that our detailed comparison with EWoK, along with the comprehensive responses to the reviewer’s questions, helps clarify the theoretical contributions, the challenges (both present and future), and the strengths of our work. We the reviewer might consider raising the final score when assessing their final evaluation.
>
> --
>
> [1] Gadot, Uri, et al. "Bring your own (non-robust) algorithm to solve robust MDPs by estimating the worst kernel." Forty-first International Conference on Machine Learning. 2024.
>
> [2] Kumar, Navdeep, et al. "Policy gradient for rectangular robust markov decision processes." Advances in Neural Information Processing Systems 36 (2023): 59477-59501.
>
> [3] Nichol, Alexander Quinn, and Prafulla Dhariwal. "Improved denoising diffusion probabilistic models." International conference on machine learning. PMLR, 2021.

---

> > ### Comment · Reviewer_2eCV · 2025-08-03
> >
> > Thank you for the response. I agree that the techniques and the idea of the work are interesting and worth exploring. But I agree with reviewer TZ4u, that the paper is ready for publication yet. I suggest that the authors to  continue exploring the direction and further submit with some more results.

---

> > > ### Author Response · Authors · 2025-08-03
> > > **Addressing new doubts and providing additional results**
> > >
> > > We thank the reviewer for the quick reply and the additional input. We believe that the results provided in the paper are more than enough to show that the performance of our algorithm is much better than several baselines (robust and not).
> > >
> > > We provided reviewer TZ4u with more results, comparing our method with an additional robust baseline (CPPO, chosen by the reviewer). The analysis highlighted once again that our method achieves superior performance both in robustness and on the nominal environment. We also provided more experiments in our last reply to reviewer TZ4u, showing that our algorithm (trained on 1.5M samples) outperforms CPPO even when trained on 6M samples. We will be happy to add these results in the final version of our manuscript.
> > >
> > > While one can always add more experiments to strengthen their claims, we strongly believe that these results, added to the ones provided in our paper, confirm the effectiveness of our method.
> > >
> > > If the reviewer would like to see more results that could help them lean towards acceptance (feasible to achieve in the last 2 days of rebuttal) we will do our best to deliver them by the August 6th deadline. Otherwise, since the previous questions have been answered in detail, we hope the reviewer will consider raising their final grade.

---

> > > > ### Author Response · Authors · 2025-08-06
> > > > **Follow-up on additional results**
> > > >
> > > > We thank the reviewer for their time and thoughtful engagement during the review process.
> > > >
> > > > As a brief follow-up, we would like to reiterate that the additional experimental results comparing AD-RRL with the CPPO baseline (as suggested by reviewer TZ4u) demonstrate that AD-RRL maintains superior performance, even when trained with significantly fewer environment interactions (1.5M for AD-RRL vs. 6M for CPPO).
> > > >
> > > > We respectfully note that reviewer TZ4u acknowledged the strength of our contributions and the effort we put into the rebuttal, and they updated their score accordingly. We hope the reviewer will consider this in their final evaluation.

---

> > > > > ### Comment · Reviewer_2eCV · 2025-08-07
> > > > >
> > > > > Thanks for the response. After going over the subsequent responses, I update my score to 4.

---

### Official Review · Reviewer_pD4N · 2025-07-01

**Clarity:** 3
**Significance:** 3
**Originality:** 3
**Rating:** 4
**Confidence:** 3

**Summary:**

This paper presents an adversarially robust reinforcement learning algorithm, termed AD-RRL, which leverages conditional value at risk (CVaR) optimization and diffusion models to generate worst-case trajectories. The paper is well-structured, and the problem being addressed is both interesting and significant. The proposed approach is theoretically sound, and the experimental results effectively demonstrate its efficacy. Overall, I believe this paper exceeds the acceptance threshold for NeurIPS.

**Questions:**

1. In the proposed method, classifier-guided diffusion is employed, built upon a classifier-free diffusion model $p_\theta(\tau_0)$. By design, diffusion models aim to replicate the historical data distribution within the dataset. Given this, how is it possible for the model to generate "trajectories that are either rare in the current environment or originate from unexplored regions of the domain," as described in line 51 of the paper? Could the authors clarify this discrepancy?
2. I can understand the approximation that $p_\theta(\tau_0\in C_\alpha|\tau_0)\approx exp(-c_0\sum\gamma^t r_t)$. Could authors provide more intuitions that why $p_\theta(\tau_i\in C_\alpha|\tau_0)\approx exp(-c_i\sum\gamma^t r_t^{(i)})$, and how to get $r_t^{(i)}$ for each $i$?
3. In Algorithm 2, $c_i$ is compute with Eq.(18), but Eq.(18) is an inequality. As stated by the authors in line 243, it should instead be calculated using Eq. (19) and set as an equality. Is this correct?

**Ethical Concerns:**

["NO or VERY MINOR ethics concerns only"]

**Final Justification:**

The authors’ rebuttal satisfactorily resolves the majority of my concerns. Consequently, I will maintain my positive assessment of this paper.

**Limitations:**

The limitations of the paper are discussed by the authors.

**Paper Formatting Concerns:**

Not at all.

**Quality:**

3

**Strengths And Weaknesses:**

**Strengths:**
1. The motivation of the paper is clearly articulated, and the content is well-written. I found the paper enjoyable to read.
2. Most of the mathematical notations are well-defined. Despite the large number of notations, they are presented in a clear and non-confusing manner.
3. This paper is theoretically sound. All theoretical components are rigorously proven or appropriately cited when sourced from other works. Additionally, all assumptions are thoroughly explained.
4. The experimental results convincingly demonstrate the effectiveness of the proposed method.


**Weaknesses:**

I did not observe any significant weaknesses in this paper.

**Minor issues:**
1. The abbreviation "RL" is introduced in line 16 but is repeated unnecessarily in line 83. Please consider removing the repetition.
2. Please ensure proper capitalization of article titles in the references. For example, "atari" should be "Atari," and "Idql" should be "IDQL."
3. In Eq. (18), the notation $r$ does not appear to represent reward information, as its definition is introduced in Appendix C. The authors should consider using a different notation to avoid potential misunderstanding and restate its definition explicitly in the main text for clarity.

---

> ### Author Rebuttal · Authors · 2025-07-31
>
> We thank the reviewer for the interesting questions and for the time dedicated to reading and evaluating our paper.
>
> - We are pleased to see that the reviewer found the paper well-structured, clearly motivated, and enjoyable to read, and appreciated the significance and theoretical soundness of our contribution.
> - We are especially grateful for the positive assessment of both the clarity of our mathematical formulation and the strength of our experimental results, as well as the overall recommendation that the paper exceeds the acceptance threshold for NeurIPS.
>
> Below, we address their questions in detail.
>
> ---
>
> **Q1. By design, diffusion models aim to replicate the historical data distribution within the dataset. Given this, how is it possible for the model to generate “trajectories that are either rare in the current environment or originate from unexplored regions of the domain,” as described in line 51 of the paper?**
>
> We thank the reviewer for highlighting this point. While diffusion models are indeed trained to replicate the distribution of historical trajectories, our approach modifies the sampling process via CVaR-based adversarial guidance, which biases the model toward trajectories associated with low return (i.e., areas where the agent tends to fail). Among low-return trajectories, we expect some to come from under-explored regions of the environment, since a low cumulative return may indicate that the agent has not learned how to behave optimally in these regions.
>
> We agree that our original phrasing may have been ambiguous and will revise the sentence in line 51 to make this distinction clearer in the final version.
>
> ---
>
> **Q2. I can understand the approximation that $p_\theta(\tau_0 \in C_\alpha \mid \tau_0) \approx \exp(-c_0 \sum \gamma^t r_t)$. Could the authors provide more intuition for why $p_\theta(\tau_i \in C_\alpha \mid \tau_i) \approx \exp(-c_i \sum \gamma^t r_t^{(i)})$, and how to obtain $r_t^{(i)}$ for each $i$?**
>
> Thank you for the question; we are happy to clarify.
>
> The reason we need this approximation at each step is as follows:
>
> - The goal is to be able to sample from $\bar p_{ \theta}(\tau_0)$. From eq. (16), and equation below line 201, we have that $\bar p_{ \theta_0}(\tau_0)\propto p_{\theta}(\tau_0)p_\theta(\tau_0\in C_\alpha|\tau_0)$.
>   - However, multiplying the distribution by the second term (a conditional probability) is usually costly and difficult. For diffusion models, the second term can be either treated as a small perturbation in each step in the diffusion process, or (often) obtained adding a multiplicative factor to the distribution in each diffusion step.
>   - Our approach consists in adding a multiplicative factor in each diffusion step (see also proof of eq. 16). Therefore, we use the same kind of approximation $p_\theta(\tau_i\in C_\alpha|\tau_i)\approx exp(-c_i\sum\gamma^t r_t^{(i)})$, so that we can exploit its smoothness for guiding the model toward adversarial rollouts.
>   - The intuition is that this form enables us to reweight trajectories at every diffusion step based on their cumulative return.
>
> - How we get $r_t^{(i)}$: note that for every generated trajectory $\tau_i$ we predict the cumulative reward $\sum_t \gamma^tr_t^{(i)}$ using a separate network, that takes as input $\tau_i$ and the diffusion step $i$. The implementation of the guidance procedure is described in section 5 of our paper (see, e.g., lines 270/271 onwards). More details on the implementation are presented in Appendix E.
>
> ---
>
> **Q3. In Algorithm 2, $c_i$ is computed with Eq. (18), but Eq. (18) is an inequality. As stated in line 243, should $c_i$ instead be calculated using Eq. (19) and set as an equality?**
>
> Thank you for pointing this out. You are correct — in Algorithm 2, $c_i$ should be computed using Eq. (19), where the value is set using an equality. However, we would like to note that using Eq. (18) would not change the computed value of $c_i$ in this setting, as the first term in the $\min$ is always greater than the second term, as shown in Proposition 4.3.
>
> We will fix this in the final version by using Eq. (19) with equality to compute $c_i$ in Algorithm 2.
>
> ---
>
> **Q4. Minor issues and notation fixes**
>
> We thank the reviewer for carefully reading the paper and noting these points. We will correct the notation and writing issues in the final version.
>
> ---
>
> We hope that, given our detailed answers to the reviewer's questions, they might consider raising the final score for our paper.

---

> > ### Comment · Reviewer_pD4N · 2025-08-06
> >
> > I appreciate the authors for their responses, which have addressed most of my questions and concerns. I will retain my positive score.

---

### Official Review · Reviewer_iLrN · 2025-07-03

**Clarity:** 3
**Significance:** 3
**Originality:** 3
**Rating:** 4
**Confidence:** 3

**Summary:**

This paper proposes a model-based reinforcement learning approach that integrates diffusion models with adversarial robustness. Specifically, the diffusion-based world model is guided to generate trajectories on which the learned policy would achieve low returns, and the policy is subsequently trained to maximize returns on these challenging trajectories. Empirical evaluations demonstrate that the proposed method outperforms baseline approaches, particularly in noisy test environments.

**Questions:**

- L206: The choice of the approximation formula seems a bit arbitrary. Can the authors clarify why that specific formula was used? For example, what happens if you remove the exponential term?
- To determine if guiding the diffusion model to generate adversarial trajectories is necessary, it would be helpful to compare the proposed method with simply introducing random noise to the diffusion model (similar to DR-PPO).

**Ethical Concerns:**

["NO or VERY MINOR ethics concerns only"]

**Final Justification:**

Most of my concerns are addressed. This paper presents a compelling approach to enhancing robustness through diffusion-based trajectory modeling. I keep my current positive rating.

**Limitations:**

.

**Paper Formatting Concerns:**

.

**Quality:**

3

**Strengths And Weaknesses:**

Strengths
- Diffusion models have recently gained widespread adoption in reinforcement learning, and this paper presents a compelling approach to enhancing robustness through diffusion-based trajectory modeling.
- The paper is well-written and easy to follow.
- Experimental results demonstrate that the proposed method matches or exceeds the performance of baseline methods, particularly when the test environment contains additional noise not encountered during training.

Weaknesses
- As noted in L51, the proposed method may tend to sample trajectories that are rare or insufficiently explored during training. This raises concerns about increased modeling errors, as the diffusion model may not have been adequately trained on such trajectories.
- The experiments are limited to relatively small-scale locomotion tasks. Evaluating the method’s scalability to more complex, high-dimensional environments would significantly strengthen the paper.
- Training guided diffusion models and computing return gradients at every diffusion step can be computationally expensive.

---

> ### Author Rebuttal · Authors · 2025-07-31
>
> We thank the reviewer for their feedback and for the time spent evaluating our paper. We are glad to see that the reviewer found the paper well-written and easy to follow, and appreciated the integration of diffusion-based trajectory modeling with adversarial robustness as a compelling and timely contribution to the field. We are also pleased that the reviewer highlighted the strength of our empirical results, noting that our method matches or outperforms strong baselines, particularly in noisy or perturbed test environments, thereby demonstrating the robustness benefits of our approach.
>
> Below, we provide detailed responses to their comments and questions.
>
> ---
>
> **Q1. The proposed method may tend to sample trajectories that are rare or insufficiently explored during training. This raises concerns about increased modeling errors, as the diffusion model may not have been adequately trained on such trajectories.**
>
> We appreciate the concern raised by the reviewer, and are happy to clarify this point.
>
> Note that our CVaR-based sampling process simply biases the diffusion process toward **trajectories with low return**, which do not need to be *rare* in general.
>
> - While diffusion models are typically trained to replicate the distribution of historical trajectories, our approach modifies the sampling process through CVaR-based adversarial guidance, biasing the model toward low-return trajectories, i.e., regions where the agent tends to fail. Among these low-return trajectories, we expect some to originate from under-explored areas of the environment, as poor cumulative return may suggest the agent has not yet learned optimal behavior in those regions. However, low returns can also arise for other reasons, such as stochastic dynamics or a sub-optimal current policy.
> - Regarding model accuracy, the training curves in the appendix show that the agent’s performance on the nominal environment continuously improves throughout training, suggesting small modeling errors.
>   - Importantly, our architecture builds on **PolyGRAD**, conditioning the diffusion process on the policy gradient. Thus, the sampled trajectories are not arbitrary outliers but are likely under the current policy, ensuring they remain realistic.
>   - The combination of adversarial guidance (AD-RRL) and policy-informed conditioning (PolyGRAD) steers sampling toward challenging yet plausible trajectories, balancing robustness and model fidelity.
>
> We will clarify these points in the final version of the manuscript.
>
> ---
>
> **Q2. The experiments are limited to relatively small-scale locomotion tasks. Evaluating the method’s scalability to more complex, high-dimensional environments would significantly strengthen the paper.**
>
> We understand the reviewer’s concern. Our current experiments focus on **well-established continuous-control benchmarks** that allow clear comparison to prior work (see for example [1,2,3,4], a small selection of papers using these same benchmarks), highlighting the advantage of our adversarial guidance mechanism. We agree that evaluating scalability to more complex, high-dimensional environments (e.g., vision-based settings) is a valuable future direction.
>
> Furthermore, our approach is modular: the adversarial guidance does not depend on domain-specific structure, suggesting that it should generalize well to more complex tasks. We aim to explore such extensions in future work.
>
> ---
>
> **Q3. Training guided diffusion models and computing return gradients at every diffusion step can be computationally expensive.**
>
> This is an important practical point, but note that our methods is approximately $1.5\times$ more computationally expensive than PolyGRAD, which does not include adversarial guidance (note that compared to PolyGrad our method requires an additional model).
>
> - Being a model-based approach, it is natural to expect it to be more expensive than model-free approaches. However, we believe this overhead is justified by the significant gains in robustness and nominal performance, as demonstrated in Figures 1–3 and Table 1.
> - AD-RRL consistently outperforms all baselines, including model-free and model-based alternatives, on both nominal environments and unobserved test-time conditions. This highlights a positive tradeoff between computational resources and robustness.
>
> We agree that reducing this computational overhead is an important direction for future work.
>
> ---
>
> **Q4. The choice of the approximation formula seems a bit arbitrary. Can the authors clarify why that specific formula was used? For example, what happens if you remove the exponential term?**
>
> We would like to highlight that we are not the first to employ such ideas. In fact, such approximation is also used in other relevant prior works (e.g., [5,6]) focusing on RL for continuous tasks.
>
> **The main idea** borrows from [5], which casts the control problem as a probabilistic inference problem. Practically speaking, they introduce a binary random variable $O_t$ indicating whether timestep $t$ is optimal, and parameterize the probability of $O_t$ as $P(O_t=1|s_t,a_t)=\exp(r(s_t,a_t))$. This is a **smooth** approximation of the indicator function $P(O_t=1|s_t,a_t)={\bf 1}_{(s_t,a_t)\in C}$, where $C$ is the set of optimal state-action pairs.
>
> - While such approximation may seem arbitrary, in fact it leads to a very natural posterior distribution $P(\tau|O_1,\dots,O_T)\propto P(\tau)\exp(\sum_t r(s_t,a_t))$. That is, the probability of observing a  trajectory (given its optimality) is given by the product between its probability to occur and the exponential of the total reward along that trajectory.
> - In AD-RRL we follow this approach, but need to adapt it to our setting: a diffusion process does not generate one state-action, but an entire trajectory at the time. For this reason, we do not model the \emph{optimality of a time-step}, but rather the optimality of an entire trajectory $\tau$:
>   - We introduce a binary random variable $O(\tau)$ indicating wether $\tau$ is optimal, and parametrize such random variable by $P(O(\tau)=1|\tau)=\exp(-c_0\sum_t \gamma^t r_t)$ (we consider the discounted return since the optimality of a trajectory is based on its total discounted return).
>   - Since this is a smooth parametrization, it enables gradient-based guidance in the diffusion process that steers sampling toward low-return (adversarial) trajectories.
>   - Without such an approximation, the adversarial objective would be **intractable** for diffusion-based planning.
>
> ---
>
> **Q5. To determine if guiding the diffusion model to generate adversarial trajectories is necessary, it would be helpful to compare the proposed method with simply introducing random noise to the diffusion model (similar to DR-PPO).**
>
> This is a great point, we thank the reviewer for the suggestion. We believe that just adding random noise to the diffusion model would not yield improvements in performance or robustness, since diffusion models inherently inject Gaussian noise at every denoising step.
>
> - As an alternative, for this rebuttal, we implemented a Domain Randomization variant of PolyGRAD (which we refer to as DR-PolyGRAD), where the agent is trained to maximize expected return over a uniform distribution of randomized environment dynamics, in line with the DR-PPO formulation.
> - Note that Domain Randomization methods assume access to the environment simulator to randomize physical parameters. This assumption is often impractical in many real-world settings where the agent does not have access to a simulator. In contrast, AD-RRL learns a dynamics model and uses it to generate entire trajectories into the future, enabling adversarial sampling without requiring additional environment access during training.
> - Due to time and resource constraints, we were able to compare the algorithms only on one environment (Hopper). Since we are not allowed to share plots over anonymized links, we provide point-wise evaluations of the two methods in the following table. We can see that AD-RRL still outperforms DR-PolyGRAD in robustness to unobserved test conditions. AD-RRL also achieves better performance than DR-PolyGRAD when the mass is close to its nominal value.
>
> | **Mass relative change** | **AD-RRL (Ours)** | **DR-PolyGRAD** |
> | --- | --- | --- |
> | 0.506 | **2431.643 ± 543.152** | **2678.814 ± 191.608** |
> | 0.749 | **3147.467 ± 12.367** | 2842.534 ± 234.759 |
> | 0.992 | **3279.055 ± 14.408** | 2952.995 ± 240.776 |
> | 1.331 | **3280.748 ± 81.662** | 2865.380 ± 17.254 |
>
> *Table 1 — Performance across mass-relative changes for Hopper (best results in bold).*
>
> ---
>
> We hope that these answers address the reviewer’s concerns and that, given the additional comparison with DR-PolyGRAD, they will consider raising their final score for our paper.
>
> ---
>
> [1] Pinto, Lerrel, et al. "Robust adversarial reinforcement learning." International conference on machine learning. PMLR, 2017.
>
> [2] Ying, Chengyang, et al. "Towards safe reinforcement learning via constraining conditional value-at-risk." arXiv preprint arXiv:2206.04436 (2022).
>
> [3] Rigter, Marc, Jun Yamada, and Ingmar Posner. "World models via policy-guided trajectory diffusion." arXiv preprint arXiv:2312.08533 (2023).
>
> [4] Zeng, Siliang, et al. "Maximum-likelihood inverse reinforcement learning with finite-time guarantees." Advances in Neural Information Processing Systems 35 (2022): 10122-10135.
>
> [5] Levine, Sergey. "Reinforcement learning and control as probabilistic inference: Tutorial and review." arXiv preprint arXiv:1805.00909 (2018).
>
> [6] Janner, Michael, et al. "Planning with diffusion for flexible behavior synthesis." arXiv preprint arXiv:2205.09991 (2022).

---

> ### Author Response · Authors · 2025-08-07
> **Follow-up on the reviewer's questions**
>
> We thank the reviewer for their positive feedback and insightful questions. In our rebuttal we have:
>
>   1. Clarified the trajectory-generation mechanism, showing how CVaR guidance steers sampling toward low-return yet policy-plausible trajectories, thereby limiting modelling error.
>   2. Explained our focus on standard continuous-control benchmarks, which enable direct comparison with prior work. Because the guidance mechanism is modular, we expect it to generalize readily to higher-dimensional tasks.
>   3. Discussed the trade-off between computational cost and performance: although AD-RRL is more expensive than some baselines, it consistently outperforms them in both nominal and perturbed conditions, which we believe to be a worth trade-off.
>   4. Clarified the exponential approximation and why a smooth, differentiable form is essential for gradient-based guidance.
>   5. Implemented the **DR-PolyGRAD** baseline you suggested; AD-RRL still outperforms it across mass variations in Hopper.
>
> If any additional clarification or experiment would be helpful before the deadline, we are happy to provide it. Otherwise, we hope these additions address your concerns and will be reflected in your final assessment.
>
> Thank you for your time and consideration.

---

### Decision · Program_Chairs · 2025-09-17

**Decision:**

Accept (poster)

**Comment:**

This paper develops a new method for robust RL with environmental uncertainty.  The objective is to learn a policy that maximizes the Conditional Value-at-Risk (CVaR) of trajectory returns—a problem previously explored in the literature. The authors build on PolyGRAD, a model-based RL method that leverages a diffusion world model and policy-guided trajectory diffusion. The main contribution of the paper is the integration of adversarial guidance into PolyGRAD to generate the worst α-percentile low return trajectories under a given policy, which are then used to iteratively improve policy robustness. Experimental results show that the proposed method, called AD-RRL, improves both the robustness and performance of policies across several MuJoCo tasks, outperforming PolyGRAD and other baseline approaches.

The reviewers noted that the paper was novel and well structured, clear to read, and addresses a significant problem. Assumptions are clearly stated and theoretical aspects are rigorous. The approach is theoretically sound, and the experiments effectively demonstrate the value of the proposed method compared to baselines, and demonstrate robustness when the test environment has additional noise not encountered during training. Overall, the paper demonstrates that a diffusion world model can be adversarially guided to incorporate robustness with theoretical guarantees, highlighting the potential of diffusion models in robust sequential decision-making.

A key limitation is computational complexity and scaling.  The experiments were for relatively small-scale locomotion tasks, and exploring scalability to more complex, high-dimensional environments would strengthen the research.  The method trains guided diffusion models and computing return gradients at every diffusion step (over entire trajectories) can be computationally expensive.  So the method as-is may be unlikely to scale to problems with high-dimensional state spaces.

The authors do compare to PPO and TRPO, and show that the proposed AD-RRL policy is more than 100x slower, so the proposed method as-is seems to be bordering on impractical.  Reviewers suggest considering recently proposed diffusion architecture alternatives such as latent diffusion models and EDM.  One reviewer suggests that “To achieve a more practical solution, it is crucial to explore these more scalable methods and adapt them to trajectory diffusion.”  This is considered reasonable and feasible.

It is also suggested by the reviewers that more comparisons are needed to existing robust RL methods as improved baselines, such as CVaR-Proximal Policy Optimization (CPPO) that has much lower complexity.

Summary: The reviewers noted the significant strengths of the paper in clarity, novelty, theoretical underpinnings, and empirical performance.  The consensus was positive, while suggesting aspects that need further work, especially exploring alternative diffusion models with lower complexity, and comparisons with state of the art robust RL methods.